# Multiple anthropogenic pressures eliminate the effects of soil microbial diversity on ecosystem functions in experimental microcosms

Gaowen Yang [1,2,3]✉, Masahiro Ryo [4,5], Julien Roy[2,3], Daniel R. Lammel[2,3], Max-Bernhard Ballhausen[2,3], Xin Jing [6], Xuefeng Zhu[7] & Matthias C. Rillig [2,3]

Biodiversity is crucial for the provision of ecosystem functions. However, ecosystems are now exposed to a rapidly growing number of anthropogenic pressures, and it remains unknown whether biodiversity can still promote ecosystem functions under multifaceted pressures. Here we investigated the effects of soil microbial diversity on soil functions and properties when faced with an increasing number of simultaneous global change factors in experimental microcosms. Higher soil microbial diversity had a positive effect on soil functions and properties when no or few (i.e., 1–4) global change factors were applied, but this positive effect was eliminated by the co-occurrence of numerous global change factors. This was attributable to the reduction of soil fungal abundance and the relative abundance of an ecological cluster of coexisting soil bacterial and fungal taxa. Our study indicates that reducing the number of anthropogenic pressures should be a goal in ecosystem management, in addition to biodiversity conservation.

[1] College of Grassland Science and Technology, China Agricultural University, Beijing 100193, China. [2] Institute of Biology, Freie Universität Berlin, Berlin 14195, Germany. [3] Berlin-Brandenburg Institute of Advanced Biodiversity Research (BBIB), Berlin 14195, Germany. [4] Leibniz Centre for Agricultural Landscape Research (ZALF), Müncheberg 15374, Germany. [5] Institute of Environmental Sciences, Brandenburg University of Technology Cottbus-Senftenberg (BTU-CS), Cottbus 03046, Germany. [6] State Key Laboratory of Grassland Agro-Ecosystems, and College of Pastoral Agriculture Science and Technology, Lanzhou University, Lanzhou, Gansu 730020, China. [7] Institute of Applied Ecology, Chinese Academy of Sciences, Shenyang 110016, China. ✉email: yanggw@cau.edu.cn

Biodiversity is fundamental for providing and sustaining ecosystem functions[1–6]. The main evidence for the positive effects of biodiversity on ecosystem functioning comes from experiments with biodiversity manipulation under ambient environmental conditions or a few anthropogenic pressures[1,4,7–9]. However, ecosystems can simultaneously encounter multiple anthropogenic pressures[10–13]. For instance, the co-occurrence of multiple pressures, or at least some combination of pressures, including nutrient eutrophication, warming, drought, mechanic compaction, heavy metal pollution, residues of plastic mulching film and pesticides, has been reported by recent studies in intensively managed agroecosystems[10,13–16]. Nevertheless, it remains unknown whether biodiversity can sustain the provisioning of ecosystem functions under multiple anthropogenic pressures.

Soil biodiversity, one of the largest reservoirs of biodiversity on Earth, is of significant importance for the maintenance of multiple ecosystem functions[17–20]. There is an urgent need to investigate how multiple anthropogenic pressures influence the effects of soil biodiversity on ecosystem functions for a better understanding of the consequence of multiple pressures on ecosystem sustainability. Previous studies suggest that a single or a combination of just a few anthropogenic pressures can regulate the effects of biodiversity on ecosystem functions through interspecific interactions[1,4,7–9]. Compared with ambient conditions, there is an even larger positive biodiversity effect on plant biomass production because of an increase in interspecific complementation in the face of a few pressures[4,9]. Moreover, single pressures, e.g., warming, have been shown to improve interspecific competition among microbes, leading to a decrease in biodiversity effects on biomass production[21].

Although single anthropogenic pressures often have limited effects on the growth of the population, the simultaneous impacts of multiple anthropogenic pressures are much more severe because of the negative and synergistic effects of multiple pressures[14,16,22]. Consequently, a decrease in the abundance of organisms can reduce ecosystem functions delivered by these species[23,24]. When population abundance is reduced to a relatively low level, the changes in ecosystem functions will depend on the abundance of the population regardless of biodiversity level[9,25]. Therefore, we hypothesize that an increasing number of anthropogenic pressures may progressively decrease the ability of soil biodiversity to promote ecosystem functions by reducing population abundance.

Here we tested this hypothesis using a microcosm experiment involving 370 micro-systems (Table 1 and Supplementary Fig. 1 and 2). This experiment had two factors: soil biodiversity (high and low soil biodiversity) and the number of global change factors (GCFs) (seven levels: 0, 1, 2, 4, 6, 8, 10) (Table 1). The dilution-to-extinction approach was used to create the high and low soil biodiversity treatments (Supplementary Methods). This approach

has reduced soil bacterial and fungal taxonomic richness by 64 and 53%, respectively, without altering their abundance in the soil inoculum (Supplementary Fig. 3). The magnitude of biodiversity loss by soil dilution in our study is similar to an extreme loss of soil biodiversity, e.g., the loss of soil fungal diversity by multiple GCFs[14]. Here we refer to soil biodiversity as the initial soil microbial diversity and community composition.

We created an increasing number of GCFs following the classic plant diversity experimental design[5,6]. The identity and composition of GCFs have been shown to influence response variables[14,16,22,26]. In the present study, we de-emphasized a focus on the identity and composition of GCFs by randomly sampling factors from a pool of ten GCFs, thus asking a simplified question primarily about the number of GCFs. Following the implementation of treatments, we measured soil respiration rate (a proxy for microbial activity), microbial abundance, enzyme activity related to nutrient cycling, physical properties (soil water-stable aggregates and water repellency), bacterial and fungal community composition and diversity after incubation.

The effects of soil biodiversity on soil functions and properties at each level of GCFs were represented by effect sizes (Hedges' g), which were calculated as the difference of the means between the high and low soil biodiversity treatments in units of the pooled standard deviation. To reveal ecological associations among soil microbes along with the increasing number of GCFs, we constructed a soil microbial network and then inferred clusters (modules) of coexisting soil bacterial and fungal taxa[27]. Our results suggest that soil microbial diversity contributes to ecosystem functions and properties only when a few GCFs are active, but that the positive diversity effects are jeopardized by the presence of multiple GCFs.

## Results

**Soil functions and properties**. We found that the high soil biodiversity treatment had a positive effect on the activity of β-D-celluliosidase, β-glucosidase and phosphatase and soil microbial activity (in terms of soil respiration) when only a few GCFs were applied, but this effect was eliminated by the co-occurrence of numerous GCFs (Fig. 1). The increasing number of GCFs reduced soil functions and properties (Supplementary Fig. 4). The microbial network comprised 223 bacterial and fungal nodes, 3490 positive edges and 300 negative edges, with a modularity of 0.364, and was clustered into four modules (Fig. 2a). Module 4 had more edges and fungal nodes, compared with other modules (Fig. 2b, c). More nodes with higher Kleinberg's hub centrality scores (represented by bigger node size) were observed in module 4 (Fig. 2a, c).

**Relationships between soil functions and properties**. Spearman correlation analysis suggested that nearly all soil functions and properties were not positively associated with bacterial and fungal richness, but were positively related to bacterial and fungal abundance and the relative abundance of microbial module 4 (Fig. 2d). These patterns were retained when data were split into the sub-data of the high and low soil biodiversity treatments (Supplementary Fig. 5). Moreover, the relative abundance of module 1 and 2 was significantly related to soil functions and properties in the high soil biodiversity treatment (Supplementary Fig. 5), but its average relative abundance was extremely low (<10%) (Supplementary Fig. 6).

The increasing number of GCFs decreased bacterial and fungal abundance in general (Fig. 2e and f). The bacterial abundance was not altered by soil biodiversity treatment (Fig. 2 h). The high soil biodiversity treatment had a positive effect on the fungal abundance in the zero and single GCF treatments, but this effect

**Table 1 Experimental design.**

| Soil biodiversity | GCFs (no.) | Replicates (no.) |
|---|---|---|
| High/low | 0 | 10 |
| High/low | 1 | 10 |
| High/low | 2 | 15 |
| High/low | 4 | 15 |
| High/low | 6 | 15 |
| High/low | 8 | 15 |
| High/low | 10 | 15 |

Note: there were 10 repeats for each single global change factor (GCF); total experimental units = (10 + 10 × 10 + 15 × 5) × 2 = 370; for the GCF levels from 2 to 10, random draws were conducted to select GCFs for each replicate.

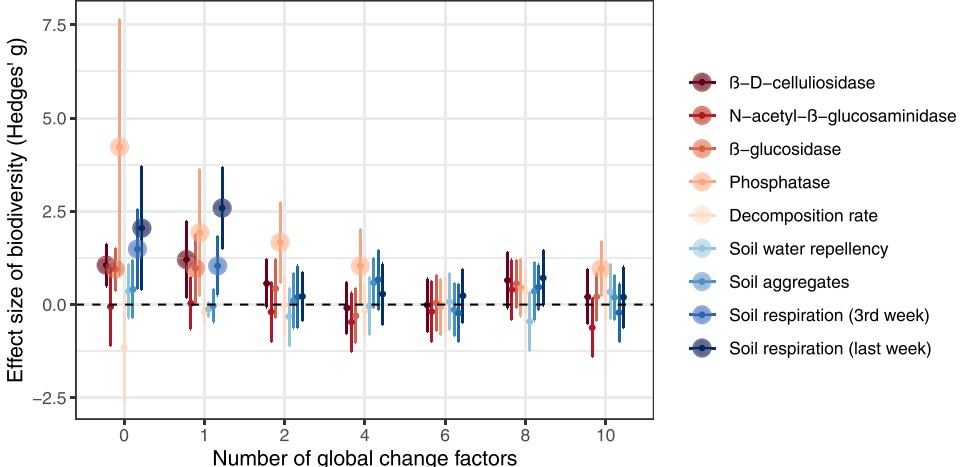

**Fig. 1 Effects of soil biodiversity treatment on soil functions and properties with an increasing number of global change factors.** Effect sizes (Hedges' g) were calculated as the difference of the means between high and low soil biodiversity treatments in units of the pooled standard deviation. Points are the mean values and the error bars are their 95% confidence intervals ($n = 10$ for the zero and single global change factor treatments; $n = 15$ for multiple global change factor treatments). The positive effect sizes without intervals overlapping zero indicate that soil biodiversity increased soil functions and properties. These effect sizes were highlighted by big points. Effect sizes with intervals overlapping zero are not significantly different from zero, meaning that soil biodiversity does not alter variables.

was diminished at a higher number of GCFs (Fig. 2i). The relative abundance of module 4 was reduced when multiple GCFs were applied, and the effect size of the high soil biodiversity treatment on the relative abundance of module 4 was suppressed when more GCFs were applied (Fig. 2g and j).

**Taxonomic and trophic composition of modules**. The increasing number of GCFs did not cause consistent changes in the overall diversity and taxonomic composition of soil bacterial and fungal communities, while soil bacterial communities clustered in the high biodiversity treatment when none and one GCF was active (Supplementary Figs. 7–9 and Supplementary Table 1). However, module 4 was composed of more fungal and bacterial species (ASVs richness) in the Mortierellomycetes, Sordariomycetes, Tremellomycetes, Chloroflexia, Gemmatimonadetes, Longimicrobia, Myxococcia, Planctomycetes (Fig. 3a), compared with other modules. In particular, most fungal species in module 4 were classified into the saprotroph group (Fig. 3b), indicating a higher ecosystem function related to decomposition. The relative abundance of these species was positively associated with soil functions (Fig. 3c), and was gradually decreased by an increasing number of GCFs (Supplementary Fig. 10).

**Effect of the number of GCFs on soil multifunctionality**. In the high soil biodiversity treatment, bacterial and fungal abundance and the relative abundance of module 4 explained 60% of the variance in the soil multifunctionality (Supplementary Fig. 11a). An increasing number of GCFs indirectly decreased soil multifunctionality via reducing the fungal abundance and the relative abundance of module 4 (Supplementary Fig. 11a). In the low soil biodiversity treatment, 45% of the variance in soil multifunctionality was mainly explained by the direct effect of an increasing number of GCFs (Supplementary Fig. 11b).

## Discussion

Our study suggests that an increasing number of GCFs eliminate the positive effects of soil biodiversity on soil functions driven by soil microbes, e.g., nutrient cycling and soil microbial activity. Past research found that biodiversity can increase ecosystem functions when faced with a single or a number of GCFs[1,4,7,28].

Accordingly, biodiversity conservation, typically focused on aboveground organisms, is regarded as one of the best ways to sustain ecosystem functions[3,6,24]. In line with these previous studies[1,4,7,28], our results show that the positive effect of soil biodiversity is maintained only when a limited number of GCFs is active, but the co-occurrence of numerous GCFs eliminates the positive biodiversity effects. Our study illustrates that reducing the number of GCFs should be an essential goal for achieving ecosystem sustainability, in addition to biodiversity conservation.

The reduction in soil biodiversity effects implies that the increasing number of GCFs could weaken the relationships between biodiversity and ecosystem functions. Past studies show that there were saturating and linear relationships between biodiversity and ecosystem functions, and these relationships are robust when faced with single or two GCFs[1,7,24]. However, the slope of these linear relationships could be reduced as a result of the reduction of biodiversity effects with the increasing number of GCFs. For saturating relationships between biodiversity and process rates, the saturation point could be reduced, because some soil functions in the high soil biodiversity treatment did not significantly differ from those in the low soil biodiversity treatment when faced with numerous GCFs. This indicates that the ability of biodiversity to provision ecosystem functions could be suppressed or eliminated when numerous GCFs are active (path 2 in Fig. 4).

Consistent with previous studies[14,16], most soil functions and properties declined with the increasing number of GCFs. The reduction in soil functions and properties, as well as soil biodiversity effects, cannot be attributed to the biodiversity loss along with the increasing number of GCFs. First, we did not observe significant positive relationships between soil functions and microbial richness (Fig. 2d). Second, there were no consistent changes in the richness and taxonomic composition of soil microbes with the increasing number of GCFs. This is in contrast with previous studies[14,15], which is likely explained by the nature of the investigated agroecosystem. In the present study, soils were collected from an intensively managed farming system. Agricultural intensification on this farmland likely has already led to an adapted microbial community consisting of soil clusters with different specialized functions and tolerance for stress. Thus, microbial diversity was tolerant to GCF treatments. However,

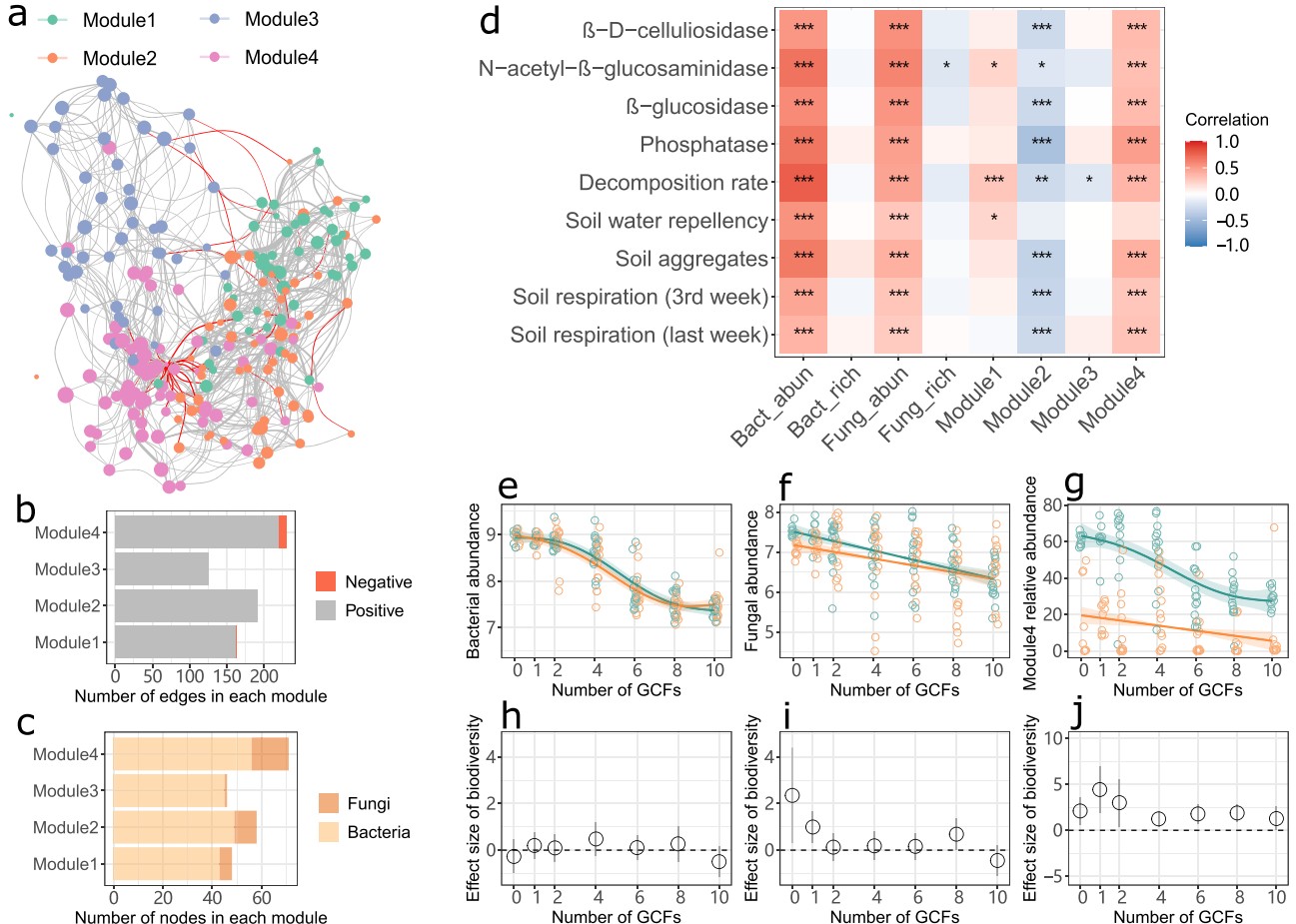

**Fig. 2 Microbial network and relationships between soil microbial variables and soil functions/properties. a** Visualization of microbial network inferred using fungal and bacterial ASV tables in all treatments. Nodes were colored according to four modules; node size represents Kleinberg's hub centrality score, indicating the connection degree of a node with adjacent nodes and the probability of being a keystone species; positive and negative edges are indicated by gray and red lines, respectively. **b, c** Number of edges and nodes (ASVs) in each module. **d** Spearman correlations between soil microbial indices and soil functions and properties. Spearman's rho statistic was employed to detect the statistical significance of correlations using a two-sided hypothesis test. The adjustment method "fdr" was used to control the false discovery rate for multiple testing correction. Positive and negative correlation coefficients are highlighted in red and blue, respectively. Asterisks represent the statistical significance of correlations ($n = 170$): ***$P < 0.001$, **$P < 0.01$, *$P < 0.05$. Note: Bact_abun, bacterial abundance; Bact_rich, bacterial ASV richness; Fung_abun, fungal abundance; Fung_rich, fungal ASV richness; Module, the relative abundance of module. **e**–**g** An increasing number of GCFs reduced bacterial and fungal abundance and the relative abundance of microbial module 4 in both high and low soil biodiversity treatments. The observed values along with the number of GCFs were fitted with generalized additive models (mean and 95% confidence intervals). The significance of the fitted models is indicated by a solid ($P < 0.05$) line ($n = 95$). The statistical test used was two-sided. **h**–**j** The effect size of soil biodiversity on bacterial and fungal abundance and the relative abundance of microbial module 4 along with an increasing number of GCFs. Effect sizes (Hedges' g) were calculated as the difference of the means between high and low soil biodiversity treatments in units of the pooled standard deviation. Points are the mean values and the error bars are their 95% confidence intervals ($n = 10$ for the zero and single global change factor treatments; $n = 15$ for multiple global change factor treatments). The positive effect sizes without intervals overlapping zero indicate that soil biodiversity increased soil functions and properties; the effect sizes with intervals overlapping zero are not significantly different from zero, meaning that soil biodiversity does not affect variables.

given that biodiversity is generally and closely associated with ecosystem functions in most studies[1–7,24], and threatened by multiple GCFs[10–13], we should not overlook the pathway by which an increasing number of GCFs can decrease ecosystem functions via biodiversity loss (path 1 in Fig. 4). Additionally, some soil processes, e.g., litter decomposition, are not sensitive to biodiversity loss, as a result of high functional redundancy from extraordinarily diverse decomposers in soils[29].

Consistent with our hypothesis, the increasing number of GCFs mainly decreased soil biodiversity effects by reducing microbial abundance (path 2 in Fig. 4). The increasing number of GCFs led to soil conditions that are detrimental to the growth of organisms, and, ultimately, decreased microbial abundance. Species abundance is often closely correlated with the functions they are

underpinning[23,24]. Thus, the reduction in microbial abundance likely caused a decline in soil functions. Agricultural intensification generally favors bacteria over fungi, indicating that fungi are more sensitive to GCF treatment than bacteria[30]. Compared with bacteria, diverse fungal communities were more critical for fungi to maintain their abundance when zero and a single GCF were applied. This likely contributed to the positive effects of soil biodiversity on soil functions when a few GCFs were applied, because fungal abundance and delivery of soil functions were positively related.

Bacterial communities experiencing none or a single GCF in the high soil biodiversity treatment were clustered tightly and separately from other GCF treatments. This indicates that bacterial community composition is relatively stable with single GCFs, which may help to maintain a positive soil biodiversity

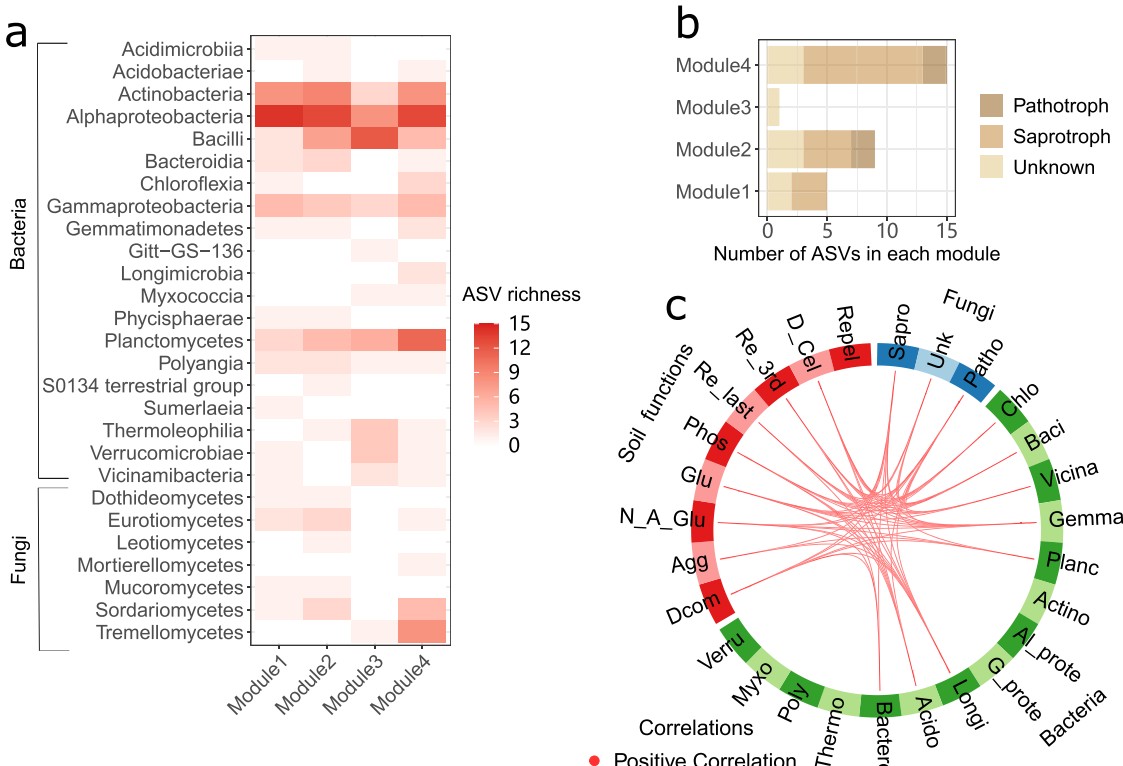

**Fig. 3 Taxonomic and trophic composition of modules. a** Taxonomic composition of modules at the class level. **b** Trophic composition of fungi in each module. Darker red indicates a greater ASV richness. **c** Visualization of correlated fungi trophic groups (blue), bacterial nodes (green) and soil functions/ properties (red). All correlations were positive, defined by Pearson's correlation $r > 0.5$ ($n = 190$). The statistical test used was two-sided. Note: Sapro, Saprotroph fungi; Patho, Pathotroph fungi; Unk, Unknown; Acido, Acidobacteriae; Actino, Actinobacteria; Al_prote, Alphaproteobacteria; Baci, Bacilli; Bactero, Bacteroidia; Chlo, Chloroflexia; G_prote, Gammaproteobacteria; Gemma, Gemmatimonadetes; Longi, Longimicrobia; Myxo, Myxococcia; Planc, Planctomycetes; Poly, Polyangia; Thermo, Thermoleophilia; Verru, Verrucomicrobiae; Vicina, Vicinamibacteria; D_Cel, β-D-celluliosidase; N_A_Glu, N-acetyl-β-glucosaminidase; Glu, β-glucosidase; Phos, phosphatase; Dcom, decomposition rate; Repel, soil water repellency; Agg, soil aggregates; Re_3rd, soil respiration (3rd week); Re_last, soil respiration (last week).

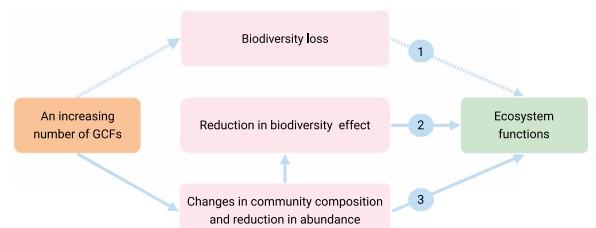

**Fig. 4 Hypothesized (dashed lines) and observed (solid lines) effects of the increasing number of global change factors (GCFs) on ecosystem functions.** An increasing number of GCFs can reduce ecosystem functions via biodiversity loss (path 1), the reduction in biodiversity effects (path 2), the changes in community composition and the reduction in abundance (path 3). The reduction in biodiversity effects could come from the changes in community composition and the reduction in population abundance induced by an increasing number of GCFs.

effect. Nevertheless, the increasing number of GCFs could reduce multifunctionality by decreasing fungal abundance (Supplementary Fig. 11a) in the high soil biodiversity treatment and by influencing community composition (path 3 in Fig. 4), e.g., reducing the relative abundance of module 4. Soil dilution partly excluded module 4 from the low soil biodiversity treatment, and therefore, module 4 did not explain the reduction in multifunctionality in the low soil biodiversity treatment. Additionally, soil dilution changed community composition shaping the

potential interactions within the communities at distinct diversity levels, which may not be captured by the network analysis. In this study, module 4 had more nodes with higher Kleinberg's hub centrality scores, indicating that the bacterial and fungal taxa in module 4 have more connections with other taxa and a higher probability of being keystone species[27].

Specifically, there were more saprophytic fungal species in module 4. The decrease in the relative abundance of these fungi could cause the reduction in soil functions related to decomposition, because fungi are a significant driver of enzyme activity[31,32]. Moreover, the relative abundance of some rare soil bacteria[33], e.g., Chloroflexia, Gemmatimonadetes, Longimicrobia, Myxococcia, Planctomycetes, was correlated with the reduction of soil functions along with an increasing number of GCFs, probably because GCFs are likely more detrimental for rare microbial species than common species[15]. Our results are consistent with previous studies showing that specific microbial taxa are important for soil functions[34,35]. The presence of productive species is a major determinant of the effect of plant diversity on productivity, known as the selection effect[36]. The changes in selection effects can affect the overall biodiversity effects[37]. Analogously, functionally important species – in our case module 4, the structuring variations of which were a main component of the whole microbial community, could be critical for a biodiversity effect. Therefore, the increasing number of GCFs could reduce the impact of soil biodiversity on soil functions by reducing the relative abundance of module 4.

In summary, our study supports the claim that soil biodiversity is essential for maintaining soil functions when faced with none or only a few GCFs. Importantly, we find that the co-occurrence of multiple GCFs can eliminate the positive effects of soil microbial diversity, probably via detrimental effects on fungal abundance and the relative abundance of a microbial ecological cluster. In addition, the reduction in soil functions with the co-occurrence of multiple GCFs could come from a decrease in soil biodiversity effects, microbial abundance and the relative abundance of a coexisting microbial cluster (Fig. 4). Our study highlights the importance of working towards reducing the number of GCFs to achieve agroecosystem sustainability. Furthermore, because multiple GCFs are increasingly threatening ecosystems worldwide[10,12,22], there is an urgent need to evaluate the consequences for biodiversity-ecosystem function relationships in different ecosystems. A better understanding of the role of multiple GCFs is crucial for ecosystem management given the challenges imposed by global change.

## Methods

**Experimental design**. This experiment was set up containing two levels of soil biodiversity (high and low soil biodiversity) and seven treatments considering the number of global change factors (GCFs) (0, 1, 2, 4, 6, 8, 10) (Table 1, Supplementary Fig. 1 and Supplementary Data 1). We used the dilution-to-extinction approach to create the high and low soil biodiversity treatments (Supplementary Methods). Soil dilution can lead to a gradual loss of rare soil microbes, which can simulate a realistic loss of soil biodiversity, because rare soil microbes are more sensitive to anthropogenic pressures, e.g., warming, nitrogen addition and drought[15]. We note that the low soil biodiversity treatment is a subset of the high biodiversity, as many rare species have been eliminated through the dilution; this approach will likely lead to relatively more tolerant microbes in the resulting communities.

An increasing number of GCFs was created inspired by the experimental design of the studies on biodiversity-ecosystem function relationships, based on random sampling from a species pool[5,6,14]. The combination of multiple GCFs was replicated 15 times at each level by randomly selecting GCFs from a pool with 10 GCFs for each replicate (Table 1 and Supplementary Data 1). For each replicate of combined GCFs, there were identical GCF combinations between the high and low soil biodiversity treatments to avoid a confounding effect of GCF combination and soil biodiversity treatments. The pool of 10 GCFs included: warming, nitrogen deposition, drought, heavy metal pollution, plastic mulching film residues, salinity, agricultural fungicide, bactericide application, surfactant contaminant and soil compaction (Supplementary Methods). These GCFs frequently occur in intensively managed agroecosystems and are treated as anthropogenic pressures[10,13–15].

**Microcosms**. This experiment was conducted using 50 ml conical Mini Bioreactors (Product Number 431720, Corning Inc., NY) as experimental units (Supplementary Fig. 2). Each Mini Bioreactor has four vents in the cap, where a hydrophobic membrane avoids microbial contamination but allows gas exchange. We filled each Mini Bioreactor with 40.0 g (dry weight, d.w.) of soil in total, which received the appropriate treatments.

**Soil sterilization and inoculum preparation**. We collected the field soil from the top 10 cm of an intensive farming system in Berlin (52.466°N, 13.303°E). Field soil was passed through a 2 mm mesh to remove large roots and stones. We sterilized 20 kg of soil for 90 min at 121 °C, and stored 2 kg of fresh soil at 4 °C. The dilution-to-extinction approach[38–42] was used to create high and low soil biodiversity (Supplementary Fig. 1). A parent inoculum suspension was prepared by mixing 100 g of fresh soil with 200 ml of sterilized VE water. The sediment settled for 1 min. The upper 200 ml of soil suspension was treated as parent inoculum suspension. 50 ml of parent inoculum suspension was added to 500 g of sterilized soil in a plastic bag, and homogenized by turning the bag up and down 30 times to obtain the inoculum of high soil biodiversity. Another 5 ml of parent inoculum suspension was mixed with 45 ml of sterilized parent inoculum suspension to create the 10⁻¹ dilution. This procedure is repeated five times to reach the 10⁻⁶ dilution. 50 ml of the 10⁻⁶ dilution was mixed and homogenized with 500 g of sterilized soil in a plastic bag to obtain the inoculum of low soil biodiversity. This whole dilution procedure was repeated five times to obtain 10 bags of soil inoculum (five bags for each soil biodiversity inoculum).

Sterile water was added to each plastic bag to reach the water content of the fresh soil in the field. All bags were closed with a sterilized cotton plug and a rubber band to avoid microbial contamination but allow gas exchange[42]. All bags were incubated in a dark room at 20 °C until similar microbial abundance was observed between the high and low soil biodiversity inoculum. Soil inoculum was homogenized by shaking and turning the bags once a week. After incubation, 2.0 g of soil in each bag was collected and stored at −80 °C for DNA extraction.

Quantitative real-time PCR (qPCR) was used to determine fungal and bacterial abundance. In the present study, it took two months to recover soil microbial biomass (Supplementary Fig. 3).

**The implementation of GCFs and harvest**. Agroecosystems, some of the most intensively managed ecosystems, are affected by the co-occurrence of multiple GCFs[13–15]. This study focused on GCFs that frequently occur in agroecosystems, including warming, nitrogen deposition, drought, heavy metal pollution, plastic film residues, salinity, agricultural fungicide and bactericide application, surfactant contaminant and soil compaction. We present the rationale for the 10 tested GCFs in the Supplementary Methods.

Loading soils were used to achieve an effective mixing of chemical agents into 40.0 g soil in each Mini Bioreactor. We created separate 'loading soil' for each GCF with chemical addition by mixing an appropriate dose of a chemical agent with sterilized soil through careful homogenization. This was done to avoid exaggerated effects of more concentrated chemicals when mixed with soil. For each chemical, 1.0 g (d.w.) of loading soil contained an appropriate dose for 40.0 g soil in a Mini Bioreactor. For instance, 1 634 mg of $NH_4NO_3$ was mixed with 100 g (d.w.) of sterilized soil, to ensure that there was about 16.34 mg of $NH_4NO_3$ in 1.0 g of sterilized loading soil. We weighed 40.0 g (d.w.) of soils, including 1.0 g (d.w.) of each loading soil, 5.0 g (d.w.) of soil inoculum (high or low soil biodiversity), an appropriate amount of film (0 or 0.16 g plastic film) and sterilized soil, according to GCF combination for each experimental unit. We put 40.0 g of mixed soils into a clean and sterilized cup (200 ml) with a cap, and then homogenized it by turning the cup up and down for 5 min using a shaking machine (Heidolph Reax 2, Heidolph Instruments GmbH & CO. KG, Schwabach, Germany). After homogenization, 40.0 g of mixed soils was transferred to a Mini Bioreactor, and a mesh bag containing about 100 mg of dry Medicago lupulina leaves (65 °C for 72 h) was buried 1 cm below the soil surface. We used a stick to press soils in each Mini Bioreactor to simulate an ambient condition (1.3 g cm⁻³) or mechanical compaction (1.7 g cm⁻³) in farmland.

For the warming treatment with an increment of 5.0 °C over the ambient temperature (20 °C), we wrapped heating cables (Exo Terra PT-2012; Hagen Deutschland GmbH & Co. KG, Holm, Germany) around the outside of the bioreactors. A set temperature was maintained by temperature controllers (Voltcraft ETC-902; Conrad Electronic SE, Hirschau, Germany), which can switch off and on heating cables depending on the real-time temperature of the outside surface of Mini Bioreactors. Mini Bioreactors were placed in beakers filled with sand to reduce thermal radiation from neighboring units. At the start of the experiment, we added suitable amounts of sterilized water to each experimental unit to reach 60% of water holding capacity for the non-drought treatment and 30% water holding capacity for the drought treatment.

All Mini Bioreactors were incubated at 20.0 °C in the dark for six weeks before the final harvest. Because there was 2.0 g of weight loss on average in each Mini Bioreactor in the first three weeks, we added 2 ml of sterilized water to each Mini Bioreactor on the first day of the fourth week. During the final harvest, soil in each Mini Bioreactor was gently homogenized using a spoon, and then 2.0 g of fresh soil was collected and stored at −80 °C for DNA extraction; 5.0 g was stored at 4 °C for the determination of soil enzyme activity; the leftover was oven-dried at 40 °C for other measurements. DNA of each soil sample was extracted from 250 mg soil, using DNeasy PowerSoil Pro Kit (QIAGEN GmbH, Hilden, Germany), following manufacturer's instructions. Soil DNA extraction was stored at −80 °C for further analysis.

**The measurement of response variables**. We measured the following response variables: microbial activity (soil respiration), microbial abundance (bacterial and fungal abundance), nutrient cycling (litter decomposition rate and soil enzyme activity), physical properties (water-stable soil aggregates and soil water repellency), bacterial and fungal biodiversity (richness and microbial network features) (See details in the Supplementary Methods). We measured soil respiration as $CO_2$ concentration (ppm h⁻¹ g⁻¹ soil) in the third and sixth week as an indicator for soil microbial activity. Bacterial and fungal abundance was estimated by qPCR. The proportional loss of litter (Medicago lupulina leaves) dry weight during soil incubation was used as an indication of decomposition rate. We measured the activity of β-glucosidase (cellulose degradation), β-D-cellulosidase (cellulose degradation), N-acetyl-β-glucosaminidase (chitin degradation) and phosphatase (organic phosphorus mineralization) using high throughput microplate assay[43,44]. A modified protocol by Kemper and Rosenau was used to measure water-stable soil aggregates[45]. Soil water repellency was measured using the water drop penetration time method[46]. High throughput sequencing (Illumina MiSeq) was used to measure the taxonomic composition of soil fungal and bacterial communities with the primers fITS7 and ITS4 for fungi and 515F-Y and 806 R for bacteria[47,48] (Supplementary Methods).

**Statistical analyses**. For diversity and community composition analysis, we excluded the samples with less than 1% of the observations of the largest sample in the ASV table. For network analysis, we then removed ASVs with low prevalence, which presented less than 20% of samples across all experimental units to reduce the high percentage of zero counts. A co-occurrence network was constructed based on both fungal and bacterial ASV tables. The *PLNnetwork* function in the *R*

package *PLNmodels* was employed to infer the network, using a sparse multivariate Poisson log-normal (PLN) model[49]. According to the Extended Bayesian Information Criterion (EBIC), the best model was extracted with the function *getBestModel*. The network was compartmentalized into different modules using the *cluster_fast_greedy* function in the *igraph* package and visualized with partial correlations with $|\rho| > 0.05$. We focused on the response of the relative abundance of modules, also known as clusters, which represent the closely associated microbes, e.g., groups of coexisting or co-evolving microbes[27]. The relative abundance of modules was calculated by summing relative abundances for individual ASVs in modules. We used the package *FUNGuildR*[50] to taxonomically parse fungal trait information, using the FUNGuild database[51].

For each single GCF treatment, we took the average response from the 10 replicates before analysis. To confirm how the effect of soil biodiversity treatment can change along with the increasing number of GCFs, we quantified the effect size of soil biodiversity treatment for each response variable using Hedges' g (mean and 95% CIs) at each level of the number of GCFs, using the R package *effsize*[52]. Hedges' g is calculated as the mean difference between the high and low soil biodiversity treatments in units of the pooled standard deviation as a paired-samples because there were identical GCFs and GCF combinations for both high and low biodiversity conditions, with the exception of the zero and 10 GCF treatments.

To evaluate how each of the response variables changes along with the number of GCFs, we applied a generalized additive model (GAM)[53]. GAM is a penalized generalized linear model that fits a nonparametric, nonlinear smooth curve[54]. The degree of smoothness of model terms is estimated as part of fitting, using the generalized cross validation. We reasonably assume that the curve shapes are different between high and low soil biodiversity treatments. Therefore, we included biodiversity conditions (low/high) as the model intercept and as the "factor smooth" smoothing class, where a smooth function is created for each factor level independently[55]. For GAM modeling, we used the *mgcv* package[55]. The dimension of the basis used to represent the smooth term was set as k = 5 so that the model does not overfit to the data. For this, we compared some other values (from 3 to 8) and confirmed that the results are essentially the same within the tested range. The other parameters were set as default.

The relationships between soil microbial indices and other soil indices were tested using Spearman correlation in the package *microbiome*, and the adjustment method "fdr" was employed to control the false discovery rate for multiple testing correction[56]. For the further multivariate integration of soil functions/properties and composition of modules, the DIABLO (Data Integration Analysis for Biomarker discovery using a Latent component method for Omics studies) was employed to detect correlation (Pearson's correlation $|r| > 0.5$) among variables using the package *mixOmics*[57].

The Z-scores for each of the eight soil functions (as shown in Fig. 1, with the exception of soil water repellency) were evaluated, and then we computed an improved weighted multifunctionality metric to represent soil multifunctionality (Supplementary Methods)[58]. Structural equation models (SEMs) were used to reveal the direct and indirect effects of an increasing number of GCFs on soil multifunctionality within each soil biodiversity treatment using the package *lavaan*[59]. We assumed that an increasing number of GCFs influences soil multifunctionality by regulating the bacterial and fungal abundance and the relative abundance of modules. All response variables were standardized to the same comparison scale using the z-score transformation before constructing SEMs. Models with optimal fitting indices were reported (Supplementary Fig. 11).

The permutational multivariate analysis (ADONIS) and non-metric multidimensional scaling (NMDS) ordination based on the Bray-Curtis distance were conducted to test the effect of soil biodiversity and GCF treatments on the community composition of bacteria and fungi using the R package *vegan*[60]. For the data handling, processing, and visualization, we used the packages *tidyverse*[61], *reshaping*[62], *cowplot*[63], *RColorBrewer*[64], *qgraph*[65], *igraph*[66], *factoextra*[67], *phyloseq*[68] and *itsadug*[69]. These data manipulation and analyses were conducted using R version 4.1.3[70]. The R script and data are available in a publicly accessible database[71].

**Reporting summary**. Further information on research design is available in the Nature Research Reporting Summary linked to this article.

## Data availability

All datasets that support the findings of this study have been deposited in figshare (https://doi.org/10.6084/m9.figshare.16988539.v6). The Silva 138.1 database used for bacteria taxonomic annotation is available in zenodo (https://zenodo.org/record/4587955#.YrZyNshfh_8). For fungal taxonomic annotation, the UNITE ITS database is available at the webpage (https://unite.ut.ee/repository.php). The FUNGuild database is available at the webpage (http://www.stbates.org/funguild_db_2.php).

## Code availability

All R script used for experimental design, data manipulation and analyses are available in figshare (https://doi.org/10.6084/m9.figshare.16988539.v6).

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

## Acknowledgements
We thank the following for help with this study: Tingting Zhao, Yun Liang and Yaqi Xu for helping with the experimental setup, Tessa Camenzind and Carlos A. Aguilar-Trigueros for measuring soil respiration, Yuqi Wu for graph art, and Mengting Maggie Yuan and Chao Liang for comments. The study was funded by the German Research Foundation (DFG Grant No. 434341960) and Chinese Universities Scientific Fund (Grant No. 1201−15052002) to G.Y. M.C.R. additionally acknowledges support from an ERC Advanced Grant (Grant No. 694368). X.Z. acknowledges support from the National Natural Science Foundation of China (Grant No. 32101382).

## Author contributions
G.Y. conceived the idea for the study. G.Y., M.R. and M.C.R. designed this experiment. G.Y. set up the experiment and conducted measurements, with the help of D.R.L. and M.B.B. in the molecular part. G.Y., M.R., J.R., X.J. and X.Z. analyzed data. G.Y. wrote the first draft, and all authors contributed to the writing of the paper.

## Competing interests
The authors declare no competing interests.
