## [Peer Review File · Nature Communications]

Reviewers' Comments:

Reviewer #1:

Remarks to the Author:

The article of Yang and coauthors investigated how applying multiple “anthropogenic pressures” on soil communities drive community assembly and functioning. The study is asking an interesting and timely question. The paper is nicely written. My major comment concerns the analysis indicating the module 4 as a major driver of ecosystem functions. It would be beneficial if authors show also to the readers the identity of those microorganisms. An example of a paper that did a good job in connecting function with network structure is: Romdhane 2022 ISME <https://doi.org/10.1038/s41396-021-01076-9>. The authors collected a really interesting dataset which in my opinion could benefit from a more in depth analysis. For example, we know that fungi is a major driver on enzymatic activity, and authors observed that fungal abundance was related to the observed enzymatic activity, but they didn't present or discuss the properties of the fungal network. In the supplemental material authors justified that “Because we observed that bacterial abundance is 10 – 100 times larger than fungal abundance only bacterial ASV table was used to construct microbial network.”. However, authors observed that fungal abundance was related to various measured ecosystem functions. More recently, PLN models have been proposed (<https://pln-team.github.io/PLNmodels>) and allow us to evaluate “merged” networks with fungi and bacterial communities which can further help us understanding how fungi and bacterial communities drive changes in ecosystem functioning. When reading the paper I thought authors could also benefit by using Picrust to infer function of the microbial community within the module 4 in comparison to other modules. This could help them further in the interpretation of the data.

Other comments:

In Figure 2 I, authors evaluate the relationship between total bacterial assembly and module 4, however, module 4 is one of the biggest modules in their dataset, therefore, it is not surprising or informative to observe that there is a positive correlation between the module 4 BNTI and the BNTI of the whole community.

Fig 2 module 4: is this module showing a higher number of nodes and edges compared to the other modules?

Measurement of microbial abundance: microbial abundance was measured in the “big soil pool” or in the individual replicates (370)? It is not very clear from the description.

Figures: Fig 1 would be a bit less crowded if the lines between global change factor levels would be deleted. There is a lot of overlapping and lines crossing each other, which don't help the reader to get the main message out of the figure.

L66: Rephrase the statement: “fewer opportunities for biological insurance”. If the insurance effect is driven by different species: fewer species able to insure the provisioning of a specific function...etc

L 69: suggestion: replace predict by hypothesize.

L72: micro-system equals replicates?

L 75: factorial design? Not sure factorial design is the best way to describe the experimental design.

L75: Rephrase this description: This experiment was set up containing two levels of soil biodiversity (high and low soil biodiversity) and seven treatments considering the number of global change factors etc...

L86-90: The experimental description is not very easy to follow. It could be beneficial for the reader if the Supplementary Table 1 would be in the main text to help on the comprehension of

this paragraph.

L72-85: Were these soils sterile in the beginning of the experiment? If yes, how?

L97: how did authors measured soil microbial activity? Using enzymatic activity and/or respiration as a proxy for activity?

L97: physical properties?

L132-133: Did authors explore what are the taxonomic composition of module 4 and module 1? It could be interesting to have a figure showing their compositions.

L120-133: module 4 and 1 refers to only bacteria or bacteria plus fungal OTUs?

L206-210: here authors suggest that module 4 harbors closely associated bacteria which are positively associated with soil functions. This statement remains very hypothetical because when looking at the Supplementary figure 9 the relative abundance of different bacterial groups is not very different between the modules 1, 2, 3 and 4. This suggests that authors could look at narrower taxonomic identity or use Picrust to infer function. Which is not a perfect approach but could help to further interpret the results here.

Reviewer #2:

Remarks to the Author:

This manuscript presents results from an experiment that examined how anthropogenic pressure changes the effect of soil biodiversity on a set of soil ecosystem functions. This is a very ambitious objective, and I am deeply impressed by all the work done by this team. However, I have a number of what I believe are major concerns:

1) Evaluating joint effects of biodiversity and combinations of anthropogenic pressures on ecosystem function is a slippery topic, with lots of potential pitfalls and conceptual challenges. For example, biodiversity, often used as an independent variable, can change during the course of the experiment. One way out of this is recognizing that in the B-EF framework, B is actually *initial* biodiversity. In that way, we can test for the effect of initial B on EF, and one can then test if effect is conditional on some anthropogenic pressure treatment (I'll call this "design 1"). This is what the current paper talks about. Conversely, one can also consider the effect of anthropogenic pressure on B, in which case initial B needs to be the same across anthropogenic pressure treatments (I'll call this "design 2"). In design 1, one can look for B changes during the course of the experiment to explain the conditionality of the B effect, but one should bear in mind that this will not necessarily give the same result as design 2, since initial B will most likely determine the effects on B during the time course of the experiment.

In my view, the intro is not effective in structuring and defining the problem, as it mixes up various concepts, and is at points rather vague about where it is going (please read carefully the first sentence, which talks about investigating "effects of B by focusing on single or two anthropogenic pressures" - not clear to me). L50-51 implies design 1, while L70-71 seems to imply design 2. L82-83 then states that the B treatment on itself represents effects from anthropogenic pressures, which is confusing. I would recommend only pitching this intro in the context of design 1, as this is what the experiment was aiming at. When mentioning effects on B, I would be careful and state that we are talking about identifying potential mechanisms explaining the environmental context dependence of the initial B effect on EF.

2) I have my reservations concerning the design:

2.1) From BEF we know that it is tricky to mistake effects of community composition for effects of B. To avoid compositional bias, people pick multiple random combinations of n species in the treatment where richness=n. What the experiment did was diluting a source community to create two levels of *initial* B: high (source community) and low (diluted community). So this means that the low B treatment is always nested within the high B treatment. Consequently, what we're

looking at is an effect of composition and not (only) of biodiversity. One way to easily solve this is by simply rephrasing initial B as initial composition, or even just refer to what it is: dilution. One other thing related to this issue: if rare microbes are more sensitive to global change (mentioned in the intro), then is it acceptable to use the dilution method (which filters out rare species) to create a low B treatment? Wouldn't this approach select for tolerant microbes in the low B treatment, thus introducing a bias?

2.2) I have the same comment concerning the anthropogenic pressures treatments. In the supplements I see there were many replicates, but I infer from the available information that these are true replicates, i.e. the same combinations of anthropogenic pressures were used within a level of the anthropogenic pressures factor. For example, the case of "2" anthropogenic pressures always represents the same combination. If this is the case, the same concern applies as in 2.1: we cannot distinguish the effect of the pressure combination from the effect of the number of pressures.

3) I wonder to what extent the effect of "number of pressures" really is an effect of the "number" as opposed to just the "total amount" of pressure. Inevitably, adding more pressures to the system will increase the net pressure the species experience. This raises the question if the results hadn't looked the same if there was just a single pressure with many "potency" levels (say 10 temperature levels all the way up to unreasonable levels). Would there be a way to somehow test for this? There are a number of relatively recent publications that can explain the effects found in this manuscript as an effect of overall potency of the pressure combinations:

<https://par.nsf.gov/servlets/purl/10073097>

<https://onlinelibrary.wiley.com/doi/full/10.1111/ele.13936>

<https://onlinelibrary.wiley.com/doi/full/10.1002/ece3.8182>

I hope these publications help thinking about how to best check again the analyses.

Detailed comments:

L34: What is "this" referring to?

L195-197: I could not understand this sentence, sorry.

L206-210: It is interesting that a given module responds more to the treatments than others. I wonder if it would be possible to explain a bit more about the functional ecology of this module, and discuss why it is affected most by the treatments?

L211: I think a species needs to be both productive and competitively superior to get a positive selection effect (cf. the literature on additive partitioning). I also don't think the sampling effect is a synonym for the selection effect. Instead, I believe the sampling effect refers to the statistical fact that throwing in more species increases the chance to get a particularly dominant one (and thus a selection effect) and/or to get a combination of complementary species (which would then cause a complementarity effect).

L278: If one works with "agroecosystems", isn't it to be expected that the "reference" conditions (no anthropogenic pressures) are actually suboptimal for the microbes?

L303: How hot was the warming treatment?

L356: Why working with means as opposed to all replicates?

L362: Is a paired test really appropriate? These do not seem to be observations made on the same experimental unit.

L369: With a GAM, I don't think one needs to assume anything about (non)linearity. In my experience, if the data suggest a linear trend, the GAM will return a linear trend. This is what's cool about GAMs.

Fig2: Panels C, E, G seem redundant with B, D, F, resp.

L589+595: I think there's an issue with the labels: B-D/E-G does not seem correct?

L592: I think a legend is missing for the biodiversity treatment.

L592-593: This sentence lacks a verb; typo I think.

Fig3: This figure is not easy to follow, and ideally it is simplified according to the followed design (design 1).

L614-615: Sound odd. Say that $EF = a + b \cdot B$, where b = the effect of B on EF (ecosystem function). What the sentence seems to be saying (if I got it right) is that a lower b can be explained from a lower B.

Reviewer #3:

Remarks to the Author:

The authors present a study looking at the combined impacts of global change factors (GCFs) on the multifunctionality of soil microorganisms. This is a very well considered manuscript that is generally written extremely well. The authors should be commended for a heroic amount of data collected. I have very few criticisms of this piece, as I think the whole idea and execution is very well thought out. I particularly admire the simplicity of the results, although a lot of extra information is given in the supplementary information.

Main queries:

Can you clarify the design of the combined GCFs? Working from the supplementary figure it is not clear how the 10 GCFs were assigned to the different microcosms. Are these randomly assigned or where they combined with particular stresses in mind? It would have been interesting if "universal" stresses and "local" stresses were mixed in some way (i.e. increasing temperature (affects all taxa) with chemical addition (affects some taxa) etc.). I appreciate that this might not add or detract much from the analysis but would be good for the reader to know how they combined. Are there plans to further analyse the "compensatory mechanisms" (line 221) as this seems to be an important line of investigation? Without this I am not sure what the co-occurrence networks are adding to the study. If two bacteria increase under a particular stress this is not indicative that they are cooperating rather they are both benefitting; the resulting interaction is unknown. When discussing "bacteria module 4" this reads like the authors are fishing for something in a huge amount of data, rather than being clear as to what is happening (difficult I grant you). Could the taxa within the module be related to whether they are rare moving to abundant in the stress and linked with increasing functionality? Can some detail be added to this result?

Point-by-point response to reviewer comments

Response to Reviewer #1

1. The article of Yang and coauthors investigated how applying multiple “anthropogenic pressures” on soil communities drive community assembly and functioning. The study is asking an interesting and timely question. The paper is nicely written. My major comment concerns the analysis indicating the module 4 as a major driver of ecosystem functions. It would be beneficial if authors show also to the readers the identity of those microorganisms. An example of a paper that did a good job in connecting function with network structure is: Romdhane 2022 ISME <https://doi.org/10.1038/s41396-021-01076-9>. The authors collected a really interesting dataset which in my opinion could benefit from a more in depth analysis. For example, we know that fungi is a major driver on enzymatic activity, and authors observed that fungal abundance was related to the observed enzymatic activity, but they didn't present or discuss the properties of the fungal network.

In the supplemental material authors justified that “Because we observed that bacterial abundance is 10 – 100 times larger than fungal abundance only bacterial ASV table was used to construct microbial network.”. However, authors observed that fungal abundance was related to various measured ecosystem functions. More recently, PLN models have been proposed (<https://pln-team.github.io/PLNmodels>) and allow us to evaluate “merged” networks with fungi and bacterial communities which can further help us understanding how fungi and bacterial communities drive changes in ecosystem functioning.

>> **Response:** Thank you very much for these positive and constructive comments.

(1) We re-constructed a merged network based on both fungal and bacterial ASV tables using PLN models. See figure 2 below and lines 351-358 (hereafter, line number was referred to the clean version manuscript) “For network analysis, we then removed ASVs with low prevalence, which presented less than 20% of samples across all experimental units to reduce the high percentage of zero counts. A co-occurrence network was constructed based on both fungal and bacterial ASV tables. The *PLNnetwork* function in the R package *PLNmodels* was employed to infer the

network, using a sparse multivariate Poisson log-normal (PLN) model (Chiquet *et al.*, 2019). According to the Extended Bayesian Information Criterion (EBIC), the best model was extracted with the function *getBestModel*.”

(2) To connect soil functions with the network structure, we use DIABLO (Data Integration Analysis for Biomarker discovery using a Latent component method for Omics studies) following Romdhane *et al.* (2022). See figure 3 below and lines 391-394 “For the further multivariate integration of soil functions/properties and composition of modules, the DIABLO (Data Integration Analysis for Biomarker discovery using a Latent component method for Omics studies) was employed to detect correlation (Pearson’s correlation $|r| > 0.5$) among variables using the package *mixOmics* (Rohart *et al.*, 2017).”.

(3) Network features and the identity of each module have been added in figure 2 and figure 3. The main results were not changed: four joint fungal-bacterial modules were detected, and the new module 4 was significantly related to most soil functions and properties.

Figure 2 Microbial network and relationships between soil microbial variables and

soil functions/properties. **(A)** Visualization of microbial network inferred using fungal and bacterial ASV tables in all treatments. Nodes were colored according to four modules; node size represents Kleinberg's hub centrality score, indicating the connection degree of a node with adjacent nodes and the probability of being a keystone species; positive and negative edges are indicated by gray and red lines, respectively. **(B-C)** Number of edges and nodes (ASVs) in each module. **(D)** Relationships between soil microbial indices and soil functions and properties. Positive and negative correlation coefficients are highlighted in red and blue, respectively. Asterisks represent the statistical significance of correlations ($n = 170$): *** $P < 0.001$, ** $P < 0.01$, * $P < 0.05$. **(E - G)** An increasing number of GCFs reduced bacterial and fungal abundance and the relative abundance of microbial module 4 in both high and low soil biodiversity treatments. The observed values along with the number of GCFs were fitted with generalized additive models (mean and 95% CIs). The significance of the fitted models is indicated by a solid ($P < 0.05$) line. **(H - J)** The effect size of soil biodiversity (Hedges' g ; mean and 95% CIs) on bacterial and fungal abundance and the relative abundance of microbial module 4 along with an increasing number of global change factors. The positive effect sizes without intervals overlapping zero indicate that soil biodiversity increased soil functions and properties; the effect sizes with intervals overlapping zero are not significantly different from zero, meaning that soil biodiversity does not affect variables.

Figure 3 Taxonomic and trophic composition of modules. **(A)** Taxonomic composition of modules at the class level. **(B)** Trophic composition of fungi in each module. Darker red indicates a higher number of ASV richness. **(C)** Visualization of correlated fungi trophic groups (blue), bacterial nodes (green) and soil functions/properties (red). All correlations were positive, defined by Pearson's correlation $r > 0.5$. Note: Sapro, Saprotroph fungi; Patho, Pathotroph fungi; Unk, Unknown; Acido, Acidobacteriae; Actino, Actinobacteria; Al_prote, Alphaproteobacteria; Baci, Bacilli; Bactero, Bacteroidia; Chlo, Chloroflexia; G_prote, Gammaproteobacteria; Gemma, Gemmatimonadetes; Longi, Longimicrobia; Myxo, Myxococcia; Planc, Planctomycetes; Poly, Polyangia; Thermo, Thermoleophilia; Verru, Verrucomicrobiae; Vicina, Vicinamibacteria; D_Cel, β -D-celluliosidase; N_A_Glu, N-acetyl-b-glucosaminidase; Glu, β -glucosidase; Phos, phosphatase; Dcom, decomposition rate; Repel, soil water repellency; Agg, soil aggregates; Re_3rd, soil respiration (3rd week); Re_last, soil respiration (last week).

- When reading the paper I thought authors could also benefit by using Picrust to infer function of the microbial community within the module 4 in comparison to other modules. This could help them further in the interpretation of the data.

>> **Response:** Thanks for these suggestions.

(1) For bacteria, we predicted the functional composition of modules using PICRUST2 with its default settings and calculated the relative abundance of microbial metabolic pathways, which were annotated based on the Kyoto Encyclopedia of Genes and Genomes (KEGG) database. However, we did not find any compositional changes in the top 20 predicted functions (PathwayL3) among modules. We then used FAPROTAX to conduct the function prediction of bacteria, but we did not observe any dramatic changes in the predicted functions (See the figure below). Thus, we did not present these results.

Figure The predicted function of bacteria in each module based on FAPROTAX

(2) Alternatively, we directly showed the taxonomic composition of modules at the Class level and the trophic composition of fungi in each module (See figure 3 in the response 1).

See lines 364-365 “We used the package FUNGuildR to taxonomically parse fungal trait information, using the FUNGuild database (Nguyen et al., 2016).”

See lines 131-139 “However, module 4 was composed of more fungal and bacterial species (ASVs richness) in the Mortierellomycetes, Sordariomycetes, Tremellomycetes, Chloroflexia, Gemmatimonadetes, Longimicrobia, Myxococcia, Planctomycetes (Fig. 3A). In particular, most fungal species in module 4 were

classified into the saprotroph group (Fig. 3B), indicating a higher ecosystem function related to decomposition. The relative abundance of these species was positively associated with soil functions (Fig. 3C), and was gradually decreased by an increasing number of GCFs (Supplementary Fig. 10).”

3. In Figure 2 I, authors evaluate the relationship between total bacterial assembly and module 4, however, module 4 is one of the biggest modules in their dataset, therefore, it is not surprising or informative to observe that there is a positive correlation between the module 4 BNTI and the BNTI of the whole community.

>> **Response:** we agree, and we have removed these results, while the identity of soil microbes in each module has been shown in figure 3.

4. Fig 2 module 4: is this module showing a higher number of nodes and edges compared to the other modules?

>> **Response:** in the new inferred network, module 4 has a slightly higher number of nodes and edges (See Fig 2B and C). This result has been added to lines 107-109 “Module 4 had more edges and fungal nodes, compared with other modules (Fig. 2B and C). More nodes with higher Kleinberg's hub centrality scores (represented by bigger node size) were observed in module 4 (Fig. 2A and C).”

5. Measurement of microbial abundance: microbial abundance was measured in the “big soil pool” or in the individual replicates (370)? It is not very clear from the description.

>> **Response:** There are two steps to measure microbial abundance. First, qPCR standards were prepared by pooling all DNA extractions at equal volumes. Second, samples were measured individually based on the standards. To avoid confusion, we divided the description into two parts, entitled “(a) *Standard preparation*” and “(b) *Sample measurement*”, respectively (See Supplementary Information).

6. Figures: Fig 1 would be a bit less crowded if the lines between global change factor levels would be deleted. There is a lot of overlapping and lines crossing each other, which don't help the reader to get the main message out of the figure.

>> **Response:** Thanks for this helpful suggestion. Figure 1 looks clearer now after removing lines (See figure 1 below).

Figure 1 Effect of soil biodiversity on soil functions and properties with an increasing number of global change factors. Effect sizes (Hedges' g; the difference of the means between high and low soil biodiversity treatments in units of the pooled standard deviation) are shown with their 95% confidence intervals (CIs). The positive effect sizes without intervals overlapping zero indicate that soil biodiversity increased soil functions and properties. These effect sizes were highlighted by big points. Effect sizes with intervals overlapping zero are not significantly different from zero, meaning that there is no effect of soil biodiversity on the variable.

7. L66: Rephrase the statement: “fewer opportunities for biological insurance”. If the insurance effect is driven by different species: fewer species able to insure the provisioning of a specific function...etc

>> **Response:** this statement has been deleted because the introduction has been rephrased according to the reviewer 2' suggestions.

8. L 69: suggestion: replace predict by hypothesize.

>> **Response:** accepted. See lines 67-69 “Therefore, we hypothesize that an increasing number of anthropogenic pressures may progressively decrease the ability of soil biodiversity to promote ecosystem functions by reducing population abundance.”

9. L72: micro-system equals replicates?

>> **Response:** yes.

10. L 75: factorial design? Not sure factorial design is the best way to describe the

experimental design.

L75: Rephrase this description: This experiment was set up containing two levels of soil biodiversity (high and low soil biodiversity) and seven treatments considering the number of global change factors etc...

>> **Response:** accepted and revised. See lines 71-73 “This experiment had two factors: soil biodiversity (high and low soil biodiversity) and the number of global change factors (GCFs) (seven levels: 0, 1, 2, 4, 6, 8, 10) (Table 1).” Furthermore, this description has been revised to show the experimental design in the Methods, in addition to Table 1 (see response 11 below).

11. L86-90: The experimental description is not very easy to follow. It could be beneficial for the reader if the Supplementary Table 1 would be in the main text to help on the comprehension of this paragraph.

>> **Response:** accepted and revised. (1) The description has been revised by pointing out the origin of this experimental design in the Methods. See line 243-249 “An increasing number of GCFs was created inspired by the experimental design of the studies on biodiversity-ecosystem function relationships, based on random sampling from a species pool (Tilman *et al.*, 1996; Huang *et al.*, 2018; Rillig *et al.*, 2019). The combination of multiple GCFs was replicated 15 times at each level by randomly selecting GCFs from a pool with 10 GCFs for each replicate (Table 1 and Supplementary Table 1). For each replicate of combined GCFs, there were identical GCF combinations between the high and low soil biodiversity treatments to avoid a confounding effect of GCF combination and soil biodiversity treatments.” To avoid repeating the statement of the method, these descriptions in detail have been moved from the introduction to the *Methods*.

(2) A supplementary table has been added in the main text to show the actual combinations of global change factors across 0, 1, 2, 4, 6, 8, and 10 GCF treatments. See Supplementary Table 1 for details.

(3) Table 1 has been added to the main text (see below).

Table 1. Experimental design.

Soil	GCFs	Replicates
biodiversity	(no.)	(no.)
High/low	0	10
High/low	1	10
High/low	2	15
High/low	4	15
High/low	6	15
High/low	8	15
High/low	10	15

Note: there were 10 repeats for each single GCF; total experimental units = $(10 + 10 \times 10 + 15 \times 5) \times 2 = 370$; for the GCF levels from 2 to 10, random draws were conducted to select GCFs for each replicate.

12. L72-85: Were these soils sterile in the beginning of the experiment? If yes, how?

>> **Response:** Yes. We sterilized soils for 90 min at 121°C before the experimental setup. See line 263 “We sterilized 20 kg of soil for 90 min at 121°C”. Each microcosm was composed of 5 g of high or low soil biodiversity inoculum and 35 g of sterilized soils.

13. L97: how did authors measured soil microbial activity? Using enzymatic activity and/or respiration as a proxy for activity?

L97: physical properties?

>> **Response:** revised. See line 84-88 “We measured soil respiration rate (a proxy for microbial activity), microbial abundance, enzyme activity related to nutrient cycling, physical properties (soil water-stable aggregates and water repellency), bacterial and fungal community composition and diversity after incubation.”

14. L132-133: Did authors explore what are the taxonomic composition of module 4 and module 1? It could be interesting to have a figure showing their compositions.

>> **Response:** Taxonomic composition of each module has now been added in Figure 3. See Figure 3 in the response 1.

15. L120-133: module 4 and 1 refers to only bacteria or bacteria plus fungal OTUs?

>> **Response:** In the previous version, only bacteria were included in modules. However, in the revised version, both bacterial and fungal sequencing data were used to build the network. Therefore, each module now has both bacterial and fungal ASVs. See Figure 2 and 3 in the response 1.

16. L206-210: here authors suggest that module 4 harbors closely associated bacteria which are positively associated with soil functions. This statement remains very hypothetical because when looking at the Supplementary figure 9 the relative abundance of different bacterial groups is not very different between the modules 1, 2, 3 and 4. This suggests that authors could look at narrower taxonomic identity or use Picrust to infer function. Which is not a perfect approach but could help to further interpret the results here.

>> **Response:** revised.

(1) The results of functional prediction using PICRUST2 and FAPROTAX were not presented because we did not find any compositional changes among modules (see the response 2). However, we now show the taxonomic composition of modules at the Class level (See Figure 3).

(2) We revised the discussion accordingly. See lines 203-210 “Specifically, there were more saprophytic fungal species in module 4. The decrease in the relative abundance of these fungi could cause the reduction in soil functions related to decomposition, because fungi are a significant driver of enzyme activity (Boer *et al.*, 2005; Soares & Rousk, 2019). Moreover, the relative abundance of some rare soil bacteria (Delgado-Baquerizo *et al.*, 2018), e.g. Chloroflexia, Gemmatimonadetes, Longimicrobia, Myxococcia, Planctomycetes, was correlated with the reduction of soil functions along with an increasing number of GCFs, probably because GCFs are likely more detrimental for rare microbial species than common species (Zhou *et al.*, 2020).”

(3) Furthermore, we reported the changes in community composition of module 4 with an increasing number of GCFs (See Supplementary Fig. 10 below).

Supplementary Fig. 10 Effects of an increasing number of global change factors (GCFs) on the community composition of module 4.

Response to Reviewer #2

1. This manuscript presents results from an experiment that examined how anthropogenic pressure changes the effect of soil biodiversity on a set of soil ecosystem functions. This is a very ambitious objective, and I am deeply impressed by all the work done by this team. However, I have a number of what I believe are major concerns: 1) Evaluating joint effects of biodiversity and combinations of anthropogenic pressures on ecosystem function is a slippery topic, with lots of potential pitfalls and conceptual challenges. For example, biodiversity, often used as an independent variable, can change during the course of the experiment. One way out of this is recognizing that in the B-EF framework, B is actually *initial* biodiversity. In that way, we can test for the effect of initial B on EF, and one can then test if effect is conditional on some anthropogenic pressure treatment (I'll call this "design 1"). This is what the current paper talks about. Conversely, one can also consider the effect of anthropogenic pressure on B, in which case initial B needs to be the same across anthropogenic pressure treatments (I'll call this "design 2"). In design 1, one can look for B changes during the course of the experiment to explain the conditionality of the B effect, but one should bear in mind that this will not necessarily give the same result as design 2, since initial B will most likely determine the effects on B during the time course of the experiment.

In my view, the intro is not effective in structuring and defining the problem, as it mixes up various concepts, and is at points rather vague about where it is going (please read carefully the first sentence, which talks about investigating "effects of B by focusing on single or two anthropogenic pressures" - not clear to me). L50-51 implies design 1, while L70-71 seems to imply design 2. L82-83 then states that the B treatment on itself represents effects from anthropogenic pressures, which is confusing. I would recommend only pitching this intro in the context of design 1, as this is what the experiment was aiming at. When mentioning effects on B, I would be careful and state that we are talking about identifying potential mechanisms explaining the environmental context dependence of the initial B effect on EF.

>> **Response:** Thank you very much for the positive comments and for pointing out

the confusion in the introduction. (1) We have rephrased the whole introduction to focus on the “design 1” and deleted the background related to the “design 2”. See lines 38-69 “Biodiversity is fundamental for providing and sustaining ecosystem functions (Tilman *et al.*, 1996; Isbell *et al.*, 2015; Craven *et al.*, 2018; Huang *et al.*, 2018; Benkwitt *et al.*, 2020; Jochum *et al.*, 2020; Loreau *et al.*, 2021). The main evidence for the positive effects of biodiversity on ecosystem functioning comes from experiments with biodiversity manipulation under ambient environmental conditions or a few anthropogenic pressures (Hautier *et al.*, 2014; Isbell *et al.*, 2015; García *et al.*, 2018; Benkwitt *et al.*, 2020; Hong *et al.*, 2021). However, ecosystems can simultaneously encounter multiple anthropogenic pressures (Orr *et al.*, 2020; Ridder *et al.*, 2020; Rillig *et al.*, 2021; Riedo *et al.*, 2021). For instance, the co-occurrence of multiple pressures, or at least some combination of pressures, including nutrient eutrophication, warming, drought, mechanic compaction, heavy metal pollution, residues of plastic mulching film and pesticides, has been reported by recent studies in intensively agroecosystems (Rillig *et al.*, 2019; Zhou *et al.*, 2020; Rillig *et al.*, 2021; Riedo *et al.*, 2021; Zandalinas *et al.*, 2021). Nevertheless, it remains unknown whether biodiversity can sustain the provisioning of ecosystem functions under multiple anthropogenic pressures.

Soil biodiversity, one of the largest reservoirs of biodiversity on Earth, is of significant importance for the maintenance of multiple ecosystem functions (Bardgett & van der Putten, 2014; Wagg *et al.*, 2014; Jing *et al.*, 2015; Delgado-Baquerizo *et al.*, 2020). There is an urgent need to investigate how multiple anthropogenic pressures influence the effects of soil biodiversity on ecosystem functions for a better understanding of the consequence of multiple pressures on ecosystem sustainability. Previous studies suggest that a single or a combination of just a few anthropogenic pressures can regulate the effect of biodiversity on ecosystem functions through interspecific interactions (Hautier *et al.*, 2014; Isbell *et al.*, 2015; García *et al.*, 2018; Benkwitt *et al.*, 2020; Hong *et al.*, 2021). Compared with ambient conditions, there was an even larger positive biodiversity effect on plant biomass production because of

an increase in interspecific complementation in the face of a few pressures (Isbell *et al.*, 2015; Hong *et al.*, 2021). Moreover, single pressure, e.g., warming, has been shown to improve interspecific competition among microbes, leading to a decrease in biodiversity effect on biomass production (Parain *et al.*, 2019).

Although single anthropogenic pressures often have limited effects on the growth of the population, the simultaneous impacts of multiple anthropogenic pressures are much more severe because of the negative and synergistic effects of multiple pressures (Rillig *et al.*, 2019; Zandalinas *et al.*, 2021; Orr *et al.*, 2022). Consequently, the decrease in the abundance of organisms can reduce the ecosystem functions delivered by these species (Tilman *et al.*, 2014; Hall *et al.*, 2018). When population abundance is reduced to a relatively low level, the changes in ecosystem functions will depend on the abundance of the population regardless of biodiversity level (Baert *et al.*, 2018; Hong *et al.*, 2021). Therefore, we hypothesize that an increasing number of anthropogenic pressures may progressively decrease the ability of soil biodiversity to promote ecosystem functions by reducing population abundance.”

2. 2) I have my reservations concerning the design:

2.1) From BEF we know that it is tricky to mistake effects of community composition for effects of B. To avoid compositional bias, people pick multiple random combinations of n species in the treatment where richness=n. What the experiment did was diluting a source community to create two levels of *initial* B: high (source community) and low (diluted community). So this means that the low B treatment is always nested within the high B treatment. Consequently, what we're looking at is an effect of composition and not (only) of biodiversity. One way to easily solve this is by simply rephrasing initial B as initial composition, or even just refer to what it is: dilution. One other thing related to this issue: if rare microbes are more sensitive to global change (mentioned in the intro), then is it acceptable to use the dilution method (which filters out rare species) to create a low B treatment? Wouldn't this approach select for tolerant microbes in the low B treatment, thus introducing a bias?

>> **Response:** accepted and revised.

(1) We have added a sentence to clarify the soil biodiversity treatment. See lines 91-92 “This study referred to soil biodiversity as the initial soil microbial diversity and community composition.”

(1) We have added the reason for the soil biodiversity setup using the dilution approach in the *Methods*. See lines 238-242 “We used the dilution-to-extinction approach to create the high and low soil biodiversity treatments (Supplementary Methods). Soil dilution can lead to a gradual loss of rare soil microbes, which can simulate a realistic loss of soil biodiversity, because rare soil microbes are more sensitive to anthropogenic pressures (Zhou *et al.*, 2020). We note that the low soil biodiversity treatment is a subset of the high biodiversity.”

(3) The dilution-to-extinction will not select tolerant microbes in the low soil biodiversity, because none of the pressures were presented during the dilution.

2.2) I have the same comment concerning the anthropogenic pressures treatments. In the supplements I see there were many replicates, but I infer from the available information that these are true replicates, i.e. the same combinations of anthropogenic pressures were used within a level of the anthropogenic pressures factor. For example, the case of “2” anthropogenic pressures always represents the same combination. If this is the case, the same concern applies as in 2.1: we cannot distinguish the effect of the pressure combination from the effect of the number of pressures.

>> **Response:** There might be some confusion about the experimental design. These have been clarified:

(1) This study focused on the increasing number of GCFs on response variables. A random sampling was conducted for each combined GCF treatment at each replicate. For instance, the 2 GCF treatment had 15 different combinations of two GCFs. Thus, there were no replicates for the combination of GCFs, and what we replicated was the number of GCFs.

(2) Table 1 was added to clarify the experimental design. In addition, we have added a supplementary table to show the actual combinations of global change factors across 0, 1, 2, 4, 6, 8 and 10 GCF treatments treatments, and to provide maximum transparency on what we did. See Supplementary Table 1 for details.

Table 1. Experimental design.

Soil	GCFs	Replicates
biodiversity	(no.)	(no.)
High/low	0	10
High/low	1	10
High/low	2	15
High/low	4	15
High/low	6	15
High/low	8	15
High/low	10	15

Note: there were 10 repeats for each single GCF; total experimental units = $(10 + 10 \times 10 + 15 \times 5) \times 2 = 370$; for the GCF levels from 2 to 10, random draws were conducted to select GCFs for each replicate.

3. 3) I wonder to what extent the effect of “number of pressures” really is an effect of the “number” as opposed to just the “total amount” of pressure. Inevitably, adding more pressures to the system will increase the net pressure the species experience. This raises the question if the results hadn’t looked the same if there was just a single pressure with many “potency” levels (say 10 temperature levels all the way up to unreasonable levels). Would there be a way to somehow test for this? There are a number of relatively recent publications that to can explain the effects found in this manuscript as an effect of overall potency of the pressure combinations:

<https://par.nsf.gov/servlets/purl/10073097>

<https://onlinelibrary.wiley.com/doi/full/10.1111/ele.13936>

<https://onlinelibrary.wiley.com/doi/full/10.1002/ece3.8182>

I hope these publications help thinking about how to best check again the analyses.

>> **Response:** Thanks for this very important and interesting comment. We have added a sentence to clarify the main target of this study. See lines 80-84 “We created an increasing number of GCFs following the classic plant diversity experimental design (Tilman *et al.*, 1996; Huang *et al.*, 2018). The identity and composition of GCFs have

been shown to influence response variables(Rillig *et al.*, 2019; Zandalinas *et al.*, 2021; Holmes *et al.*, 2021; Orr *et al.*, 2022). In the present study, we de-emphasized a focus on the identity and composition of GCFs by randomly sampling factors from a pool of ten GCFs, thus asking a simplified question primarily about the number of GCFs.”

We agree that part of the effect of factor number is also that there were additional materials and energy added across the factor gradient. However, given the nature of these factors, these cannot be simply added up, since we have added energy (warming), resources, non-resource abiotic factors, toxins and others.

Increasing the number of pressures is associated with some aspects not only an increase in the total amount of pressures but also the variability of pressure effects, and the number of possible pressure interactions (Rillig *et al.*, 2019). This is also the reality of global change occurring in the environment. This study did not intend to disentangle these concurrent mechanisms, and there is no method developed for making separation possible experimentally. Controlling for overall addition amounts (stressor intensification) is realistic when only a few types of drivers and/or a few types of response variables are considered (such as different types of microplastic that affect the population growth rate of a species, for example). Yet, we here target several response variables, and thus there is no single way to fix the "total amount of pressure while controlling stressor richness. Moreover, there were no true replicates for each GCF combination (See response 2).

Additionally, not all GCFs exerted “pressures” in the sense of decreasing process rates, when looked at individually. For instance, compared with control, we found that bactericide application increased the activity of phosphatase and N-acetyl- β -glucosaminidase in the high soil biodiversity treatment (See the figure below); N deposition and warming dramatically promoted decomposition rate in our previous study (Rillig *et al.*, 2019).

Figure The effect of soil biodiversity treatment on soil enzyme activity for each GCF.

Detailed comments:

4. L34: What is “this” referring to?

>> **Response:** this sentence has been deleted from abstract.

5. L195-197: I could not understand this sentence, sorry.

>> **Response:** revised. See lines 189-191 “Compared with bacteria, diverse fungal communities were more critical for fungi to maintain their abundance when zero and a single GCF were applied.”

6. L206-210: It is interesting that a given module responds more to the treatments than others. I wonder if it would be possible to explain a bit more about the functional ecology of this module, and discuss why it is affected most by the treatments?

>> **Response:** revised. We have re-constructed the network using both bacterial and fungal ASV table. See response 1 to reviewer 1 for details. Figure 3 has been added to show the compositional and functional differences among modules (See Figure 3 below).

(1) See the results in lines 131-139 “However, module 4 was composed of more fungal and bacterial species (ASVs richness) in the Mortierellomycetes, Sordariomycetes, Tremellomycetes, Chloroflexia, Gemmatimonadetes, Longimicrobia, Myxococcia, Planctomycetes (Fig. 3A). In particular, most fungal species in module 4 were classified into the saprotroph group (Fig. 3B), indicating a higher ecosystem function related to decomposition. The relative abundance of these species was positively

associated with soil functions (Fig. 3C), and was gradually decreased by an increasing number of GCFs (Supplementary Fig. 10).”

Figure 3 Taxonomic and trophic composition of modules. **(A)** Taxonomic composition of modules at the class level. **(B)** Trophic composition of fungi in each module. Darker red indicates a higher number of ASV richness. **(C)** Visualization of correlated fungi trophic groups (blue), bacterial nodes (green) and soil functions/properties (red). All correlations were positive, defined by Pearson’s correlation $r > 0.5$. Note: Sapro, Saprotroph fungi; Patho, Pathotroph fungi; Unk, Unknown; Acido, Acidobacteriae; Actino, Actinobacteria; Al_prote, Alphaproteobacteria; Baci, Bacilli; Bactero, Bacteroidia; Chlo, Chloroflexia; G_prote, Gammaproteobacteria; Gemma, Gemmatimonadetes; Longi, Longimicrobia; Myxo, Myxococcia; Planc, Planctomycetes; Poly, Polyangia; Thermo, Thermoleophilia; Verru, Verrucomicrobiae; Vicina, Vicinamibacteria; D_Cel, β -D-cellulosidase; N_A_Glu, N-acetyl-b-glucosaminidase; Glu, β -glucosidase; Phos, phosphatase; Dcom, decomposition rate; Repel, soil water repellency; Agg, soil aggregates; Re_3rd, soil respiration (3rd week); Re_last, soil respiration (last week).

(2) See the discussion in lines 203-210 “Specifically, there were more saprophytic fungal species in module 4. The decrease in the relative abundance of these fungi

could cause the reduction in soil functions related to decomposition, because fungi are a significant driver of enzyme activity (Boer *et al.*, 2005; Soares & Rousk, 2019). Moreover, the relative abundance of some rare soil bacteria (Delgado-Baquerizo *et al.*, 2018), e.g. Chloroflexia, Gemmatimonadetes, Longimicrobia, Myxococcia, Planctomycetes, was correlated with the reduction of soil functions along with an increasing number of GCFs, probably because GCFs are likely more detrimental for rare microbial species than common species (Zhou *et al.*, 2020).”

7. L211: I think a species needs to be both productive and competitively superior to get a positive selection effect (cf. the literature on additive partitioning). I also don't think the sampling effect is a synonym for the selection effect. Instead, I believe the sampling effect refers to the statistical fact that throwing in more species increases the chance to get a particularly dominant one (and thus a selection effect) and/or to get a combination of complementary species (which would then cause a complementarity effect).

>> **Response:** accepted and revised. We have removed the sampling effect. See lines 211-213 “The presence of productive species is a major determinant of the effect of plant diversity on productivity, known as selection effect (Loreau & Hector, 2001).”

8. L278: If one works with “agroecosystems”, isn't it to be expected that the “reference” conditions (no anthropogenic pressures) are actually suboptimal for the microbes?

>> **Response:** yes. As expected, higher soil functions/properties were detected in the zero GCF treatment. Since the soil came from an agricultural field in the first place, it is not really the case that these microbes had experienced no anthropogenic pressures. See lines 171-175 “In the present study, soils were collected from an intensively managed farming system. Agricultural intensification on this farmland likely has already been selected for an adapted microbial community consisting of soil clusters with different specialized functions and tolerance for stress.”

9. L303: How hot was the warming treatment?

>> **Response:** 25°C was set up for the warming treatment, which is an increment of 5.0 °C over the ambient temperature. We used 20°C as the ambient temperature, because it is the average soil temperature (0-10 cm) from May to September during the past three

years in the sampling site. We clarified the temperature in line 312 “For the warming treatment with an increment of 5.0 °C over the ambient temperature (20°C),...”

10. L356: Why working with means as opposed to all replicates?

>> **Response:** In our study, each single GCF treatment is analogous to the monoculture in the plant diversity experiment. Because we used different combinations of GCFs as true replicate for the combined GCF treatment, the 10 repeats of each single GCF treatment was treated as pseudo-replicates. Therefore, we computed the means of each single GCF treatment to avoid pseudoreplication.

11. L362: Is a paired test really appropriate? These do not seem to be observations made on the same experimental unit.

>> **Response:** Yes. A paired test is appropriate for our study. For each combined GCF treatment, there were 15 different combinations of multiple GCFs. For each GCF combination (replicate) at each GCF level, there were identical GCF combinations between both high and low soil biodiversity treatments to avoid a confounding effect of GCF combination and soil biodiversity treatments. Thus, a paired test was used to detect the difference between the high and low soil biodiversity treatments. However, we used a normal test for the zero and 10 GCF treatments. See lines 371-375 “Hedges’ g is calculated as the mean difference between the high and low soil biodiversity treatments in units of the pooled standard deviation as a paired-samples t-test because there were identical GCFs and GCF combinations for both high and low biodiversity conditions, with the exception of the zero and 10 GCF treatments.”

12. L369: With a GAM, I don’t think one needs to assume anything about (non)linearity. In my experience, if the data suggest a linear trend, the GAM will return a linear trend. This is what’s cool about GAMs.

>> **Response:** accepted and revised. We removed the nonlinear assumption. See lines 380-381 “We reasonably assume that the curve shapes are different between high and low soil biodiversity treatments.”

13. Fig2: Panels C, E, G seem redundant with B, D, F, resp.

>> **Response:** In the revised version, Figure 2E, F and G reported the changes in responses along with an increasing number of GCF, while Figure 2H, I and J showed

the effect of soil biodiversity on these responses.

14. L589+595: I think there's an issue with the labels: B-D/E-G does not seem correct?

L592: I think a legend is missing for the biodiversity treatment.

>> **Response:** Revised. The order of the labels and the figure legend have been revised in Figure 2 (See Fig 2 below).

Figure 2 Microbial network and relationships between soil microbial variables and soil functions/properties.

15. L592-593: This sentence lacks a verb; typo I think.

>> **Response:** revised. See the legend of figure 2 in line 629-630: “The observed values along with the number of GCFs were fitted with generalized additive models (mean and 95% CIs).”

16. Fig3: This figure is not easy to follow, and ideally it is simplified according to the followed design (design 1).

L614-615: Sound odd. Say that $EF = a+b*B$, where b =the effect of B on EF (ecosystem function). What the sentence seems to be saying (if I got it right) is that a lower b can be explained from a lower B .

>> **Response:** accepted and revised.

(1) We deleted the lines between biodiversity loss and reduction in biodiversity effect, and revised the legend of Figure 4. See Figure 4 below.

Figure 4 Hypothesized (dashed lines) and observed (solid lines) effects of the increasing number of global change factors (GCFs) on ecosystem functions. An increasing number of GCFs can reduce ecosystem functions via biodiversity loss (path 1), the reduction in biodiversity effect (path 2), the changes in community composition and the reduction in abundance (path 3).

(2) The present study focused on the changes in biodiversity effect along with an increasing number of GCFs. Our results can also advance our understanding of how multiple GCFs affect ecosystem functions, as summarized in Figure 4. To clarify the important implications, a summary sentence has been added to lines 223-226 “In addition, the reduction in soil functions under the co-occurrence of multiple GCFs could come from the decrease in soil biodiversity effect, microbial abundance and the relative abundance of a coexisting microbial cluster (Fig. 4).”

Response to Reviewer #3

1. The authors present a study looking at the combined impacts of global change factors (GCFs) on the multifunctionality of soil microorganisms. This is a very well considered manuscript that is generally written extremely well. The authors should be commended for a heroic amount of data collected. I have very few criticisms of this piece, as I think the whole idea and execution is very well thought out. I particularly admire the simplicity of the results, although a lot of extra information is given in the supplementary information.

>> **Response:** Thanks very much for these positive comments. We have improved this manuscript based on the comments. See details below.

2. Main queries: Can you clarify the design of the combined GCFs? Working from the supplementary figure it is not clear how the 10 GCFs were assigned to the different microcosms. Are these randomly assigned or were they combined with particular stresses in mind? It would have been interesting if “universal” stresses and “local” stresses were mixed in some way (i.e. increasing temperature (affects all taxa) with chemical addition (affects some taxa) etc.). I appreciate that this might not add or detract much from the analysis but would be good for the reader to know how they combined.

>> **Response:** accepted and revised. We have clarified the experimental design in this revision.

(1) Table 1 was added to clarify the experimental design (See below). The experimental design has been rephrased in the *Methods*.

See lines 235-253 “This experiment was set up containing two levels of soil biodiversity (high and low soil biodiversity) and seven treatments considering the number of global change factors (GCFs) (0, 1, 2, 4, 6, 8, 10) (Table 1, Supplementary Fig. 1 and Supplementary Table 1). We used the dilution-to-extinction approach to create the high and low soil biodiversity treatments (Supplementary Methods). Soil dilution can lead to a gradual loss of rare soil microbes, which can simulate a realistic loss of soil biodiversity, because rare soil microbes are more sensitive to

anthropogenic pressures (Zhou *et al.*, 2020). We note that the low soil biodiversity treatment is a subset of the high biodiversity.

An increasing number of GCFs was created inspired by the experimental design of the studies on biodiversity-ecosystem function relationships, based on random sampling from a species pool (Tilman *et al.*, 1996; Huang *et al.*, 2018; Rillig *et al.*, 2019). The combination of multiple GCFs was replicated 15 times at each level by randomly selecting GCFs from a pool with 10 GCFs for each replicate (Table 1 and Supplementary Table 1). For each replicate of combined GCFs, there were identical GCF combinations between the high and low soil biodiversity treatments to avoid a confounding effect of GCF combination and soil biodiversity treatments. The pool of 10 GCFs included: warming, nitrogen deposition, drought, heavy metal pollution, plastic mulching film residues, salinity, agricultural fungicide, bactericide application, surfactant contaminant and soil compaction. These GCFs frequently occur in intensively managed agroecosystems and are treated as anthropogenic pressures (Rillig *et al.*, 2019; Zhou *et al.*, 2020; Rillig *et al.*, 2021; Riedo *et al.*, 2021).”

Table 1. Experimental design.

Soil biodiversity	GCFs (no.)	Replicates (no.)
High/low	0	10
High/low	1	10
High/low	2	15
High/low	4	15
High/low	6	15
High/low	8	15
High/low	10	15

Note: there were 10 repeats for each single GCF; total experimental units = $(10 + 10 \times 10 + 15 \times 5) \times 2 = 370$; for the GCF levels from 2 to 10, random draws were conducted to select GCFs for each replicate.

(2) All combinations of GCF treatment came from random sampling of ten GCF pool.

None of GCF was assigned or combined with particular stresses. In general, the present design cannot be used to test the effects of particular GCF combinations. We additionally provide a table now that also contains all the actual GCF combinations which were chosen by random draws in the GCF combinations treatments from 2 to 10 (See Supplementary Table 1 for details).

3. Are there plans to further analyse the “compensatory mechanisms” (line 221) as this seems to be an important line of investigation? Without this I am not sure what the co-occurrence networks are adding to the study. If two bacteria increase under a particular stress this is not indicative that they are cooperating rather they are both benefitting; the resulting interaction is unknown. When discussing “bacteria module 4” this reads like the authors are fishing for something in a huge amount of data, rather than being clear as to what is happening (difficult I grant you). Could the taxa within the module be related to whether they are rare moving to abundant in the stress and linked with increasing functionality? Can some detail be added to this result?

>> **Response:** Thank you very much for these supportive comments. We have made some edits. See below:

(1) The sequencing data provide the compositional changes of soil microbes. The compensatory dynamics of soil microbes cannot be measured based on the sequencing data in this study. Therefore, we deleted the “compensatory” concept from this manuscript and rephrased the introduction. See lines 49-69 “Soil biodiversity, one of the largest reservoirs of biodiversity on Earth, is of significant importance for the maintenance of multiple ecosystem functions (Bardgett & van der Putten, 2014; Wagg *et al.*, 2014; Jing *et al.*, 2015; Delgado-Baquerizo *et al.*, 2020). There is an urgent need to investigate how multiple anthropogenic pressures influence the effects of soil biodiversity on ecosystem functions for a better understanding of the consequence of multiple pressures on ecosystem sustainability. Previous studies suggest that a single or a combination of just a few anthropogenic pressures can regulate the effect of biodiversity on ecosystem functions through interspecific interactions (Hautier *et al.*, 2014; Isbell *et al.*, 2015; García *et al.*, 2018; Benkwitt *et al.*, 2020; Hong *et al.*, 2021). Compared with ambient conditions, there was an even larger positive biodiversity

effect on plant biomass production because of an increase in interspecific complementation in the face of a few pressures (Isbell *et al.*, 2015; Hong *et al.*, 2021). Moreover, single pressure, e.g., warming, has been shown to improve interspecific competition among microbes, leading to a decrease in biodiversity effect on biomass production (Parain *et al.*, 2019).

Although single anthropogenic pressures often have limited effects on the growth of the population, the simultaneous impacts of multiple anthropogenic pressures are much more severe because of the negative and synergistic effects of multiple pressures (Rillig *et al.*, 2019; Zandalinas *et al.*, 2021; Orr *et al.*, 2022). Consequently, the decrease in the abundance of organisms can reduce the ecosystem functions delivered by these species (Tilman *et al.*, 2014; Hall *et al.*, 2018). When population abundance is reduced to a relatively low level, the changes in ecosystem functions will depend on the abundance of the population regardless of biodiversity level (Baert *et al.*, 2018; Hong *et al.*, 2021). Therefore, we hypothesize that an increasing number of anthropogenic pressures may progressively decrease the ability of soil biodiversity to promote ecosystem functions by reducing population abundance.”

(2) In the revised manuscript, the network was reconstructed using both bacterial and fungal sequencing data. We found that the new module 4 was closely and positively related to most soil functions/properties. Module composition at the class level was added to Figure 3A (See below). Interestingly, there were more saprotroph fungi (Figure 3B) and some rare bacteria (Figure 3C) were associated with soil functions/properties. These species in module 4 might be important for provisioning soil functions.

(3) See the results of module 4 in lines 131-139 “However, module 4 was composed of more fungal and bacterial species (ASVs richness) in the Mortierellomycetes, Sordariomycetes, Tremellomycetes, Chloroflexia, Gemmatimonadetes, Longimicrobia, Myxococcia, Planctomycetes (Fig. 3A). In particular, most fungal species in module 4 were classified into the saprotroph group (Fig. 3B), indicating a higher ecosystem function related to decomposition. The relative abundance of these

species was positively associated with soil functions (Fig. 3C), and was gradually decreased by an increasing number of GCFs (Supplementary Fig. 10).”

(4) See the discussion of module 4 in lines 203-210 “Specifically, there were more saprophytic fungal species in module 4. The decrease in the relative abundance of these fungi could cause the reduction in soil functions related to decomposition, because fungi are a significant diver of enzyme activity (Boer *et al.*, 2005; Soares & Rousk, 2019). Moreover, the relative abundance of some rare soil bacteria (Delgado-Baquerizo *et al.*, 2018), e.g. Chloroflexia, Gemmatimonadetes, Longimicrobia, Myxococcia, Planctomycetes, was correlated with the reduction of soil functions along with an increasing number of GCFs, probably because GCFs are likely more detrimental for rare microbial species than common species (Zhou *et al.*, 2020).”

Figure 3 Taxonomic and trophic composition of modules. (A) Taxonomic composition of modules at the class level. (B) Trophic composition of fungi in each module. Darker red indicates a higher number of ASV richness. (C) Visualization of correlated fungi trophic groups (blue), bacterial nodes (green) and soil functions/properties (red). All correlations were positive, defined by Pearson’s correlation $r > 0.5$. Note: Sapro, Saprotroph fungi; Patho, Pathotroph fungi; Unk,

Unknown; Acido, Acidobacteriae; Actino, Actinobacteria; Al_prote, Alphaproteobacteria; Baci, Bacilli; Bactero, Bacteroidia; Chlo, Chloroflexia; G_prote, Gammaproteobacteria; Gemma, Gemmatimonadetes; Longi, Longimicrobia; Myxo, Myxococcia; Planc, Planctomycetes; Poly, Polyangia; Thermo, Thermoleophilia; Verru, Verrucomicrobiae; Vicina, Vicinamibacteria; D_Cel, β -D-cellulosidase; N_A_Glu, N-acetyl-b-glucosaminidase; Glu, β -glucosidase; Phos, phosphatase; Dcom, decomposition rate; Repel, soil water repellency; Agg, soil aggregates; Re_3rd, soil respiration (3rd week); Re_last, soil respiration (last week).

Supplementary Fig. 10 Effects of increasing number of global change factors (GCFs) on community composition of module 4.

References

- Baert, J.M., Eisenhauer, N., Janssen, C.R. & De Laender, F. (2018). Biodiversity effects on ecosystem functioning respond unimodally to environmental stress. *Ecol. Lett.*, 21, 1191-1199.
- Bardgett, R.D. & van der Putten, W.H. (2014). Belowground biodiversity and ecosystem functioning.

Nature, 515, 505-511.

- Benkwitt, C.E., Wilson, S.K. & Graham, N.A.J. (2020). Biodiversity increases ecosystem functions despite multiple stressors on coral reefs. *Nat. Ecol. Evol.*, 4, 919–926.
- Boer, W.d., Folman, L.B., Summerbell, R.C. & Boddy, L. (2005). Living in a fungal world: impact of fungi on soil bacterial niche development. *FEMS Microbiol. Rev.*, 29, 795-811.
- Chiquet, J., Robin, S. & Mariadassou, M. (2019). Variational inference for sparse network reconstruction from count data. In: *Proceedings of the 36th International Conference on Machine Learning* (eds. Kamalika, C & Ruslan, S). PMLR Proceedings of Machine Learning Research, pp. 1162-1171.
- Craven, D., Eisenhauer, N., Pearse, W.D., Hautier, Y., Isbell, F., Roscher, C. *et al.* (2018). Multiple facets of biodiversity drive the diversity–stability relationship. *Nat. Ecol. Evol.*, 2, 1579-1587.
- Delgado-Baquerizo, M., Oliverio, A.M., Brewer, T.E., Benavent-González, A., Eldridge, D.J., Bardgett, R.D. *et al.* (2018). A global atlas of the dominant bacteria found in soil. *Science*, 359, 320-325.
- Delgado-Baquerizo, M., Reich, P.B., Trivedi, C., Eldridge, D.J., Abades, S., Alfaro, F.D. *et al.* (2020). Multiple elements of soil biodiversity drive ecosystem functions across biomes. *Nat. Ecol. Evol.*, 4, 210-220.
- García, F.C., Bestion, E., Warfield, R. & Yvon-Durocher, G. (2018). Changes in temperature alter the relationship between biodiversity and ecosystem functioning. *Proc. Natl. Acad. Sci. USA*, 115, 10989-10994.
- Hall, E.K., Bernhardt, E.S., Bier, R.L., Bradford, M.A., Boot, C.M., Cotner, J.B. *et al.* (2018). Understanding how microbiomes influence the systems they inhabit. *Nat. Microbiol.*, 3, 977-982.
- Hautier, Y., Seabloom, E.W., Borer, E.T., Adler, P.B., Harpole, W.S., Hillebrand, H. *et al.* (2014). Eutrophication weakens stabilizing effects of diversity in natural grasslands. *Nature*, 508, 521-525.
- Holmes, M., Spaak, J.W. & De Laender, F. (2021). Stressor richness intensifies productivity loss but mitigates biodiversity loss. *Ecol Evol*, 11, 14977-14987.
- Hong, P., Schmid, B., De Laender, F., Eisenhauer, N., Zhang, X., Chen, H. *et al.* (2021). Biodiversity promotes ecosystem functioning despite environmental change. *Ecol. Lett.*, 25, 555-569.
- Huang, Y., Chen, Y., Castro-Izaguirre, N., Baruffol, M., Brezzi, M., Lang, A.N. *et al.* (2018). Impacts of species richness on productivity in a large-scale subtropical forest experiment. *Science*, 362, 80-83.
- Isbell, F., Craven, D., Connolly, J., Loreau, M., Schmid, B., Beierkuhnlein, C. *et al.* (2015). Biodiversity increases the resistance of ecosystem productivity to climate extremes. *Nature*, 526, 574-U263.
- Jing, X., Sanders, N.J., Shi, Y., Chu, H., Classen, A.T., Zhao, K. *et al.* (2015). The links between ecosystem multifunctionality and above- and belowground biodiversity are mediated by climate. *Nat. Commun.*, 6, 8159.
- Jochum, M., Fischer, M., Isbell, F., Roscher, C., van der Plas, F., Boch, S. *et al.* (2020). The results of biodiversity–ecosystem functioning experiments are realistic. *Nat. Ecol. Evol.*, 4, 1485–1494.
- Loreau, M. & Hector, A. (2001). Partitioning selection and complementarity in biodiversity experiments. *Nature*, 412, 72-76.
- Loreau, M., Barbier, M., Filotas, E., Gravel, D., Isbell, F., Miller, S.J. *et al.* (2021). Biodiversity as

- insurance: from concept to measurement and application. *Biol. Rev.*, 96, 2333-2354.
- Nguyen, N.H., Song, Z., Bates, S.T., Branco, S., Tedersoo, L., Menke, J. *et al.* (2016). FUNGuild: An open annotation tool for parsing fungal community datasets by ecological guild. *Fungal Ecol.*, 20, 241-248.
- Orr, J.A., Vinebrooke, R.D., Jackson, M.C., Kroeker, K.J., Kordas, R.L., Mantyka-Pringle, C. *et al.* (2020). Towards a unified study of multiple stressors: divisions and common goals across research disciplines. *P Roy Soc B-Biol Sci*, 287, 20200421.
- Orr, J.A., Luijckx, P., Arnoldi, J.-F., Jackson, A.L. & Piggott, J.J. (2022). Rapid evolution generates synergism between multiple stressors: linking theory and an evolution experiment. *Glob. Change Biol.*, 28, 1740-1752.
- Parain, E.C., Rohr, R.P., Gray, S.M. & Bersier, L.F. (2019). Increased temperature disrupts the biodiversity-ecosystem functioning relationship. *Am. Nat.*, 193, 227-239.
- Ridder, N.N., Pitman, A.J., Westra, S., Ukkola, A., Hong, X.D., Bador, M. *et al.* (2020). Global hotspots for the occurrence of compound events. *Nat. Commun.*, 11, 5956.
- Riedo, J., Wettstein, F.E., Rösch, A., Herzog, C., Banerjee, S., Büchi, L. *et al.* (2021). Widespread occurrence of pesticides in organically managed agricultural soils—the ghost of a conventional agricultural past? *Environ. Sci. Technol.*, 55, 2919-2928.
- Rillig, M.C., Ryo, M., Lehmann, A., Aguilar-Trigueros, C.A., Buchert, S., Wulf, A. *et al.* (2019). The role of multiple global change factors in driving soil functions and microbial biodiversity. *Science*, 366, 886-890.
- Rillig, M.C., Ryo, M. & Lehmann, A. (2021). Classifying human influences on terrestrial ecosystems. *Glob. Change Biol.*, 27, 2273-2278.
- Rohart, F., Gautier, B., Singh, A. & Lê Cao, K.-A. (2017). mixOmics: An R package for 'omics feature selection and multiple data integration. In: *PLoS Comp. Biol.*, p. e1005752.
- Romdhane, S., Spor, A., Aubert, J., Bru, D., Breuil, M.-C., Hallin, S. *et al.* (2022). Unraveling negative biotic interactions determining soil microbial community assembly and functioning. *ISME J.*, 16, 296-306.
- Soares, M. & Rousk, J. (2019). Microbial growth and carbon use efficiency in soil: Links to fungal-bacterial dominance, SOC-quality and stoichiometry. *Soil Biol. Biochem.*, 131, 195-205.
- Tilman, D., Wedin, D. & Knops, J. (1996). Productivity and sustainability influenced by biodiversity in grassland ecosystems. *Nature*, 379, 718-720.
- Tilman, D., Isbell, F. & Cowles, J.M. (2014). Biodiversity and ecosystem functioning. *Annu. Rev. Ecol. Evol. Syst.*, 45, 471-493.
- Wagg, C., Bender, S.F., Widmer, F. & van der Heijden, M.G.A. (2014). Soil biodiversity and soil community composition determine ecosystem multifunctionality. *Proc. Natl. Acad. Sci. USA*, 111, 5266-5270.
- Zandalinas, S.I., Sengupta, S., Fritschi, F.B., Azad, R.K., Nechushtai, R. & Mittler, R. (2021). The impact of multifactorial stress combination on plant growth and survival. *New Phytol.*, 230, 1034-1048.
- Zhou, Z., Wang, C. & Luo, Y. (2020). Meta-analysis of the impacts of global change factors on soil microbial diversity and functionality. *Nat. Commun.*, 11, 3072.

Reviewers' Comments:

Reviewer #1:

Remarks to the Author:

Yang and co-authors significantly improved their manuscript incorporating the reviewers' comments. However, I have one major concern for the manuscript now. I apologize to the authors for not noticing this in the previous round of revisions. Figure 2D shows that module 2 is strongly negatively related to the functions measured while module 4 is positively related to the functions. At the same time the ordinations of microbial community composition (Supplementary Fig. 7) show a very clear differentiation between the high and low biodiversity treatments. Therefore, it is very likely that the differences in the overall network are mainly due to community composition in the high versus low diversity microcosms and not due to different interactions between the microorganisms. It would be more appropriate if authors perform two distinct networks one for the high diversity treatment and another one for the low diversity treatment as these treatments strongly shaped microbial composition. Doing the networks separately would allow authors to evaluate how the 10 GCFs studied here impacted the microbial interactions within each diversity treatment. Authors do have a big number of replicates (370) that would allow to perform such analysis.

I would also suggest that figure 4 could be replaced by an actual structural equation model (SEM) in which authors test those hypothesized paths to understand how these drivers might be influencing ecosystem functions directly and/or indirectly. When using a SEM authors can extract the information from their networks (degree of connectivity, number of nodes, etc) and input into the SEM to evaluate if changes in interactions or if mainly microbial abundance is responsible for the decrease in functioning. As the results are presented right now they suggest that module 4 is autocorrelated to bacterial and fungal abundance and it is unclear if module 4 drives or not EFs. Thus, I would suggest that authors need to re-evaluate their network analysis and to perform a SEM before accepting the manuscript.

Minor comments:

L35: soil bacterial and bacterial taxa?

L209: driver

Reviewer #2:

Remarks to the Author:

I congratulate the authors on their thorough revisions and constructive attitude in their rebuttal letter. I agree with most of their answers to my concerns. However, I still feel uncomfortable talking about the dilution treatment creating a "diversity" gradient. Yes, of course, dilution tends to reduce diversity, but also changes composition (see my original comment). While I admit "biodiversity effects" sounds cooler than "dilution effects", I truly don't think replacing the former with the latter would weaken the paper (much to the contrary actually). Further, I was confused by the response just before my point 3.3 (if rare microbes are more sensitive, then won't the dilution method make low biodiverse communities less sensitive as it filters out rare species). The response is that "the dilution-to-extinction will not select tolerant microbes in the low soil biodiversity, because none of the pressures were presented during the dilution." But then I wonder, what is meant by the "rare species are more sensitive" statement in the intro? Sensitive to what then? I reckon to other global change factors than the ones used in the experiment? If so, specify which ones please. In Figure 4, finally, it is not clear what arrows are part of paths 2 vs. 3.

Minor comment: L175: delete 'been'

Reviewer #3:

Remarks to the Author:

The authors have fully addressed my queries with the manuscript.

Point-by-point response to reviewer comments

Response to Reviewer #1

1. Yang and co-authors significantly improved their manuscript incorporating the reviewers' comments. However, I have one major concern for the manuscript now. I apologize to the authors for not noticing this in the previous round of revisions. Figure 2D shows that module 2 is strongly negatively related to the functions measured while module 4 is positively related to the functions. At the same time the ordinations of microbial community composition (Supplementary Fig. 7) show a very clear differentiation between the high and low biodiversity treatments. Therefore, it is very likely that the differences in the overall network are mainly due to community composition in the high versus low diversity microcosms and not due to different interactions between the microorganisms. It would be more appropriate if authors perform two distinct networks one for the high diversity treatment and another one for the low diversity treatment as these treatments strongly shaped microbial composition. Doing the networks separately would allow authors to evaluate how the 10 GCFs studied here impacted the microbial interactions within each diversity treatment. Authors do have a big number of replicates (370) that would allow to perform such analysis.

>> **Response:** Thank you very much for these comments.

(1) We completely understand what the reviewer means, and we also considered this analysis, since it is potentially very interesting. However, even though we overall have a high number of replicates, as the reviewer points out correctly, there are still not enough replicates to infer reliable networks separately. If we aim to evaluate an increasing number of GCF on microbial interactions within each diversity treatment, network analyses should be performed at each GCF treatment within each soil biodiversity treatment. This will lead to at least 14 networks (2 levels of diversity treatment \times 7 levels of GCFs). Besides, if we construct two networks within each soil biodiversity treatment, we can only obtain two groups of network features, which is not enough to measure the effect of GCF treatment on network features.

We have already done these network analyses in a previous version of this study,

which was not reported in this version. Although networks could be visualized (See Figure R1 below), some network features, e.g., degree of connectivity, betweenness, and closeness, cannot be computed because of poor fit. *In general, 10 or 15 replicates (or even fewer replicates after removing taxa with low abundance) for each network (table 1) are not reliable for computing networks in this study.* In addition, we did not observe a consistent increase or decrease in network features which can be computed, e.g., node number (also see the bacterial and fungal richness in Supplementary Fig. 9). Therefore, the networks in Figure R1 below were not added to this manuscript.

Figure R1 Network visualization and their features at each GCF treatment within each soil biodiversity treatment.

Alternatively, cohesion analysis was conducted to reveal positive and negative associations among microbial taxa (see Supplementary Fig. 14 below), following Herren and McMahon (2017). This approach can calculate cohesion values for each sample, which allow us to test the effect of treatments on these values. However, the results of cohesion analysis indicated that neither the increasing number of GCFs nor soil biodiversity treatment caused consistent changes in the positive and negative cohesion (Supplementary Fig. 14).

See lines 357-365 in the Supplementary Method: “To further investigate microbial

interactions and quantify the connectivity of microbial communities, we calculated cohesion values, following Herren and McMahon (2017). Cohesion value provides insights into associations among microbial taxa caused by both positive and negative species interactions. The cohesion value represents the strength and number of positive and negative associations. Microbial communities with greater negative cohesion and less positive cohesion are more stable (Hernandez *et al.*, 2021). The results of cohesion analysis indicated that neither the increasing number of GCFs nor soil biodiversity treatment caused consistent changes in positive and negative cohesion (Supplementary Fig. 14).”

Supplementary Fig. 14 Positive and negative cohesions along an increasing number of GCFs within each soil biodiversity treatment.

(2) In the previous revision, we have removed the concept of insurance effect (microbial interaction) from the whole manuscript and rephrased the introduction, following Reviewer 2’s suggestion. The current study did not focus on the interaction among soil microbes, and we did not observe any consistent changes in microbial interactions (Supplementary Fig. 14). In this study, most microbes were simultaneously and dramatically reduced by an increasing number of GCFs (Figure 2E and F). This is because multiple GCFs had a synergistic (strong negative) effect on response variables (Rillig *et al.*, 2021; Zandalinas & Mittler, 2022). As a result, we detected numerous positive but minor negative correlations in Fig 2A.

(3) We focused on the network, including all bacterial and fungal taxa. The idea of constructing a whole (global) network was to extract community assemblies without a priori assumptions but based on co-occurrences, which has been widely used in

previous studies (Delgado-Baquerizo *et al.*, 2018; Fan *et al.*, 2021; Zhang *et al.*, 2022; Romdhane *et al.*, 2022; Wang *et al.*, 2022). This approach can detect the difference in microbial community composition between the high and low soil biodiversity treatment. For instance, these taxa from different modules captured the changes in microbial community composition, and the relative abundance of module 4 is strongly correlated with soil multifunctionality in the high soil biodiversity treatment, while this correlation was not observed in the low soil biodiversity treatment (see Supplementary Fig. 11 in the next response). Furthermore, we have made edits for the explanation of the module 2 response. See lines 116-119 “Moreover, the relative abundance of module 1 and 2 was significantly related to soil functions and properties in the high soil biodiversity treatment (Supplementary Fig. 5), but its average relative abundance was extremely low (< 10%) (Supplementary Fig. 6).”

2. I would also suggest that figure 4 could be replaced by an actual structural equation model (SEM) in which authors test those hypothesized paths to understand how these drivers might be influencing ecosystem functions directly and/or indirectly. When using a SEM authors can extract the information from their networks (degree of connectivity, number of nodes, etc) and input into the SEM to evaluate if changes in interactions or if mainly microbial abundance is responsible for the decrease in functioning. As the results are presented right now they suggest that module 4 is autocorrelated to bacterial and fungal abundance and it is unclear if module 4 drives or not EFs.

Thus, I would suggest that authors need to re-evaluate their network analysis and to perform a SEM before accepting the manuscript.

>> **Response:** accepted and revised.

(1) SEMs were performed to show how an increasing number of GCFs influences soil functions indicated by the multifunctionality index. Supplementary Fig. 11 was added to the supplementary information (see below).

See lines 141 -148 in the results: “Effect of the number of GCFs on soil multifunctionality. In the high soil biodiversity treatment, bacterial and fungal abundance and the relative abundance of module 4 explained 60% of the variance in soil multifunctionality (Supplementary Fig. 11A). An increasing number of GCFs

indirectly decreased soil multifunctionality via reducing fungal abundance and the relative abundance of module 4 (Supplementary Fig. 11A). In the low soil biodiversity treatment, 45% of the variance in soil multifunctionality was mainly explained by the direct effect of an increasing number of GCFs (Supplementary Fig. 11B)."

See lines 206-209 in the discussion "Nevertheless, the increasing number of GCFs could reduce soil functions by decreasing fungal abundance (Supplementary Fig. 11A) and influencing community composition (path 3 in Fig. 4), e.g., reducing the relative abundance of module 4."

See lines 408-418 in the method "The Z-scores for each of the eight soil functions (as shown in Fig. 1, with the exception of soil water repellency) were evaluated, and then we computed an improved weighted multifunctionality metric to represent soil multifunctionality (Supplementary Methods) (Manning *et al.*, 2018). Structural equation models (SEMs) were used to reveal the direct and indirect effects of an increasing number of GCFs on soil multifunctionality within each soil biodiversity treatment using the package *lavaan* (Rosseel, 2012). We assumed that an increasing number of GCFs influences soil multifunctionality by regulating the bacterial and fungal abundance and the relative abundance of modules. All response variables were standardized to the same comparison scale using the z-score transformation before constructing SEMs. Models with optimal fitting indices were reported (Supplementary Fig. 11)."

Supplementary Fig. 11 Structural equation models illustrating the direct and indirect effects of an increasing number of GCFs on soil multifunctionality in the high (A) and low (B) soil biodiversity treatments. Solid orange arrows represent positive paths ($P < 0.05$), solid blue arrows represent negative paths ($P < 0.05$), and grey dashed arrows indicate non-significant paths ($P > 0.05$). Numbers adjacent to arrows are the standardized path coefficients. Arrow width is scaled to the magnitude of the standardized path coefficients. R^2 indicates the proportion of variance explained by predictors. The goodness-of-fit statistics are shown at the bottom of each model: χ^2 and

P values of test statistics, degrees of freedom, comparative fit index (CFI), root mean square error of approximation (RMSEA), and standardized root mean square residual (SRMR).

We added a section in the Supplementary Methods to show the method of quantifying ecosystem multifunctionality. See lines 366-387 in the Supplementary Methods “We adopted an averaging approach to quantify ecosystem multifunctionality (Jing *et al.*, 2020) using eight out of 16 response variables (see the above section for the corresponding functions/properties represented by each response variable). These response variables were soil respiration, litter decomposition rate, the activity of β -glucosidase (cellulose degradation), β -D-celluliosidase (cellulose degradation), N-acetyl-b-glucosaminidase (chitin degradation) and phosphatase (organic phosphorus mineralization), and water-stable soil aggregates. We standardized all response variables to the same comparison scale using the z-score transformation. Before the standardization, soil respiration was reflected using the form of $-fi + \max(fi)$ (Byrnes *et al.*, 2014), because a low value of soil respiration is considered as a desirable ecosystem function. We calculated an unweighted multifunctionality metric and a weighted multifunctionality metric. The former was the mean values of the eight standardized response variables. The latter was calculated using the methods presented by Manning *et al.* (2018). Specifically, a cluster analysis was conducted to produce a dendrogram tree using hierarchical agglomerative clustering. The optimal number of clusters was determined using the Elbow method (Kassambara & Mundt, 2020). Individual functions were weighted based on five clusters (i.e. 0.5 for the activity of β -glucosidase and β -D-celluliosidase, 0.5 for litter decomposition rate and N-acetyl-b-glucosaminidase, 0.5 for the first and the last respiration, and 1 for the remaining response variables). The weighted multifunctionality metric was calculated by taking the mean values of the all the weighted individual response variables. We reported results for the weighted multifunctionality metric because it improved the normality of data distribution.”

(2) While we completely see the reviewer’s point, and have carried out the interesting analysis as suggested, we feel that figure 4 should not be replaced by these actual SEMs,

since the figure makes a broader point. Figure 4 summarizes the present and previous studies and also is intended to inspire further studies. The results of SEMs can only support path 3 in Fig. 4, i.e. that an increasing number of GCFs influences soil functions by regulating microbial community composition and abundance. We cannot directly test whether an increasing number of GCFs influenced soil functions by reducing soil biodiversity (path 1) and biodiversity effects (path 2), because there were only two levels of soil biodiversity treatment.

(3) Network features were not included in SEM, as those networks are not reliable. Cohesion values did not show a consistent change along with an increasing number of GCFs, and thus, this index was not used for constructing SEM. See response 1 for details.

3. Minor comments:

L35: soil bacterial and bacterial taxa?

L209: driver

>> **Response:** revised. See lines 32-34 “This was attributable to the reduction of soil fungal abundance and the relative abundance of an ecological cluster of coexisting soil bacterial and fungal taxa.”

And lines 213-215 “The decrease in the relative abundance of these fungi could cause the reduction in soil functions related to decomposition, because fungi are a significant driver of enzyme activity.”

Response to Reviewer #2

1. I congratulate the authors on their thorough revisions and constructive attitude in their rebuttal letter. I agree with most of their answers to my concerns. However, I still feel uncomfortable talking about the dilution treatment creating a “diversity” gradient. Yes, of course, dilution tends to reduce diversity, but also changes composition (see my original comment). While I admit “biodiversity effects” sounds cooler than “dilution effects”, I truly don’t think replacing the former with the latter would weaken the paper (much to the contrary actually).

>> **Response:** Thank you very much for these thoughtful comments. We actually completely agree with the reviewer; however, we feel that ‘dilution’ refers more to the method used to achieve a decrease in biodiversity, and we would rather keep the word ‘biodiversity’ as a circumscription of the topic we wish to address. To avoid confusion in the manuscript itself, we have clarified as follows, following Reviewer 2’s suggestion in the previous version, to avoid the confusion of the biodiversity effect, in lines 91-92 “This study referred to soil biodiversity as the initial soil microbial diversity and community composition.”

2. Further, I was confused by the response just before my point 3.3 (if rare microbes are more sensitive, then won’t the dilution method make low biodiverse communities less sensitive as it filters out rare species). The response is that “the dilution-to-extinction will not select tolerant microbes in the low soil biodiversity, because none of the pressures were presented during the dilution.” But then I wonder, what is meant by the “rare species are more sensitive” statement in the intro? Sensitive to what then? I reckon to other global change factors than the ones used in the experiment? If so, specify which ones please.

>> **Response:** Thank you so much for pointing this out.

(1) Reviewer 2 is right. The dilution-to-extinction approach can filter out rare species, some of which are likely also sensitive to GCFs. As a result, the remaining community of microbes may contain members that are more tolerant to GCFs. However, this is not too dissimilar to what is happening in the natural ecosystem. For instance, GCFs will

lead to the loss of rare species (Zhou *et al.*, 2020), and simultaneously, there will be relatively more tolerant species in microbial communities following GCFs. Nevertheless, both situations were simulated by the dilution approach. In addition, there were similar response trends of both bacterial and fungal abundance along with GCF treatments (Fig 2E and F), indicating that microbes in both low and high soil biodiversity treatments were equally affected by multiple GCFs. Therefore, tolerant microbes in the low soil biodiversity treatment may not introduce a bias (Reviewer 2's concern in the previous revision). To alert the reader to this point, a sentence has been added in the method section in lines 252-255 "We note that the low soil biodiversity treatment is a subset of the high biodiversity, as many rare species have been eliminated through the dilution; this approach will likely lead to relatively more tolerant microbes in the resulting communities."

(2) We have specified GCF of a previous study in the method section in lines 249-252 "Soil dilution can lead to a gradual loss of rare soil microbes, which can simulate a realistic loss of soil biodiversity, because rare soil microbes are more sensitive to anthropogenic pressures, e.g., warming, nitrogen addition and drought (Zhou et al., 2020)."

3. In Figure 4, finally, it is not clear what arrows are part of paths 2 vs. 3.

>> **Response:** Thank you; revised. An additional explanation has been added in the figure legend. See lines 688-690 "The reduction in biodiversity effect could come from the changes in community composition and the reduction in population abundance induced by an increasing number of GCFs."

4. Minor comment: L175: delete 'been'.

>> **Response:** Thank you; revised. See lines 181-184 "Agricultural intensification on this farmland likely has already led to an adapted microbial community consisting of soil clusters with different specialized functions and tolerance for stress."

Response to Reviewer #3

1. The authors have fully addressed my queries with the manuscript.

>> **Response:** Thank you very much for your constructive comments.

References

- Byrnes, J.E.K., Gamfeldt, L., Isbell, F., Lefcheck, J.S., Griffin, J.N., Hector, A. *et al.* (2014). Investigating the relationship between biodiversity and ecosystem multifunctionality: challenges and solutions. *Methods Ecol. Evol.*, 5, 111-124.
- Delgado-Baquerizo, M., Oliverio, A.M., Brewer, T.E., Benavent-González, A., Eldridge, D.J., Bardgett, R.D. *et al.* (2018). A global atlas of the dominant bacteria found in soil. *Science*, 359, 320-325.
- Fan, K., Delgado-Baquerizo, M., Guo, X., Wang, D., Zhu, Y.-g. & Chu, H. (2021). Biodiversity of keystone phylotypes determines crop production in a 4-decade fertilization experiment. *ISME J.*, 15, 550-561.
- Hernandez, D.J., David, A.S., Menges, E.S., Searcy, C.A. & Afkhami, M.E. (2021). Environmental stress destabilizes microbial networks. *ISME J.*
- Herren, C.M. & McMahon, K.D. (2017). Cohesion: a method for quantifying the connectivity of microbial communities. *ISME J.*, 11, 2426-2438.
- Jing, X., Prager, C.M., Classen, A.T., Maestre, F.T., He, J.-S. & Sanders, N.J. (2020). Variation in the methods leads to variation in the interpretation of biodiversity-ecosystem multifunctionality relationships. *Journal of Plant Ecology*, 13, 431-441.
- Kassambara, A. & Mundt, F. (2020). *factoextra*: Extract and visualize the results of multivariate data analyses. R package version 1.0.7. <https://CRAN.R-project.org/package=factoextra>.
- Manning, P., van der Plas, F., Soliveres, S., Allan, E., Maestre, F.T., Mace, G. *et al.* (2018). Redefining ecosystem multifunctionality. *Nat. Ecol. Evol.*, 2, 427-436.
- Rillig, M.C., Lehmann, A., Orr, J.A. & Waldman, W.R. (2021). Mechanisms underpinning non-additivity of global change factor effects in the plant-soil system. *New Phytol.*, 232, 1535-1539.
- Romdhane, S., Spor, A., Aubert, J., Bru, D., Breuil, M.-C., Hallin, S. *et al.* (2022). Unraveling negative biotic interactions determining soil microbial community assembly and functioning. *ISME J.*, 16, 296-306.
- Rosseel, Y. (2012). *lavaan*: An R package for structural equation modeling. *J. Stat. Softw.*, 48, 1 - 36.
- Wang, Y.-F., Chen, P., Wang, F.-H., Han, W.-X., Qiao, M., Dong, W.-X. *et al.* (2022). The ecological clusters of soil organisms drive the ecosystem multifunctionality under long-term fertilization. *Environ. Int.*, 161, 107133.
- Zandalinas, S.I. & Mittler, R. (2022). Plant responses to multifactorial stress combination. *New Phytol.*, n/a.
- Zhang, J., Dolfing, J., Liu, W., Chen, R., Zhang, J., Lin, X. *et al.* (2022). Beyond the snapshot: identification of the timeless, enduring indicator microbiome informing soil fertility and crop production in alkaline soils. *Environmental Microbiome*, 17, 25.
- Zhou, Z., Wang, C. & Luo, Y. (2020). Meta-analysis of the impacts of global change factors on soil microbial diversity and functionality. *Nat. Commun.*, 11, 3072.

Reviewers' Comments:

Reviewer #1:

Remarks to the Author:

I appreciate that Yang and co-authors tried to carefully answer to the concerns I raised in the previous revision round. I consider that the manuscript improved with the revisions. Their results suggest that diversity manipulation (dilution approach) was the major driver of community composition (Supplementary Figure 7). This creates some constraints with generating a global network and to infer drivers of ecosystem functioning from the network (as a significant fraction of community members have been excluded from half of the samples). The results from the structural equation model (SEM) also suggest the relevance of diversity manipulation for shaping the community. In the high soil biodiversity SEM the relative abundance of module 4 was well explained ($R^2 = 0.47$), however within the low soil biodiversity SEM a much lower percentage of variance regarding module 4 has been explained by the model ($R^2 = 0.08$) suggesting that this module was not present very likely due to the diversity manipulation. One of the major points in this paper is that module 4 might be a major driver of ecosystem functioning which is reduced with GCFs. However, if the diversity manipulation excluded module 4 from the low diversity treatment it becomes difficult to make further conclusions. If authors chose to make a single global network I would consider important that they make a statement to clarify for the reader the limitation of the analysis when discussing the network results. To acknowledge that diversity manipulation was also a driver of community composition shaping the potential interactions within the communities at distinct diversity levels.

Point-by-point response to reviewer comments

Response to Reviewer #1

1. I appreciate that Yang and co-authors tried to carefully answer to the concerns I raised in the previous revision round. I consider that the manuscript improved with the revisions. Their results suggest that diversity manipulation (dilution approach) was the major driver of community composition (Supplementary Figure 7). This creates some constraints with generating a global network and to infer drivers of ecosystem functioning from the network (as a significant fraction of community members have been excluded from half of the samples). The results from the structural equation model (SEM) also suggest the relevance of diversity manipulation for shaping the community. In the high soil biodiversity SEM the relative abundance of module 4 was well explained ($R^2 = 0.47$), however within the low soil biodiversity SEM a much lower percentage of variance regarding module 4 has been explained by the model ($R^2 = 0.08$) suggesting that this module was not present very likely due to the diversity manipulation. One of the major points in this paper is that module 4 might be a major driver of ecosystem functioning which is reduced with GCFs. However, if the diversity manipulation excluded module 4 from the low diversity treatment it becomes difficult to make further conclusions. If authors chose to make a single global network I would consider important that they make a statement to clarify for the reader the limitation of the analysis when discussing the network results. To acknowledge that diversity manipulation was also a driver of community composition shaping the potential interactions within the communities at distinct diversity levels.

>> **Response:** accepted and revised. Thank you very much for these comments. We agree that the diversity manipulation (the dilution-to-extinction approach) not only reduced soil microbial diversity but also changed the composition of soil microbes. Especially, module 4 was reduced or even excluded during the diversity manipulation. We have specified the SEM accordingly, and have added a statement to that effect in the discussion and the legend of SEM figure 11 in the Supplementary Information. See lines 206-215 in the main text: “Nevertheless, the increasing number of GCFs could reduce multifunctionality by decreasing fungal abundance (Supplementary Fig. 11A)

in the high soil biodiversity treatment and by influencing community composition (path 3 in Fig. 4), e.g., reducing the relative abundance of module 4. Soil dilution partly excluded module 4 from the low soil biodiversity treatment, and therefore, module 4 did not explain the reduction in multifunctionality in the low soil biodiversity treatment. Additionally, soil dilution changed community composition shaping the potential interactions within the communities at distinct diversity levels, which may not be captured by the network analysis.”

See the legend of Supplementary Fig. 11: “Soil dilution has diminished module 4, and thus this module likely did not explain multifunctionality in the low-diversity treatment.”